# Disentangling the relationship between cancer mortality and COVID-19 in the US

Chelsea L Hansen[1,2,3]*, Cécile Viboud[1], Lone Simonsen[1,2]

[1]Division of International Epidemiology and Population Studies, Fogarty International Center, National Institutes of Health, Bethesda, United States; [2]PandemiX Center, Dept of Science & Environment, Roskilde University, Roskilde, Denmark; [3]Brotman Baty Institute, University of Washington, Seattle, United States

*For correspondence:
chelsea.hansen@nih.gov

Competing interest: The authors declare that no competing interests exist.

**Abstract** Cancer is considered a risk factor for COVID-19 mortality, yet several countries have reported that deaths with a primary code of cancer remained within historic levels during the COVID-19 pandemic. Here, we further elucidate the relationship between cancer mortality and COVID-19 on a population level in the US. We compared pandemic-related mortality patterns from underlying and multiple cause (MC) death data for six types of cancer, diabetes, and Alzheimer's. Any pandemic-related changes in coding practices should be eliminated by study of MC data. Nationally in 2020, MC cancer mortality rose by only 3% over a pre-pandemic baseline, corresponding to ~13,600 excess deaths. Mortality elevation was measurably higher for less deadly cancers (breast, colorectal, and hematological, 2–7%) than cancers with a poor survival rate (lung and pancreatic, 0–1%). In comparison, there was substantial elevation in MC deaths from diabetes (37%) and Alzheimer's (19%). To understand these differences, we simulated the expected excess mortality for each condition using COVID-19 attack rates, life expectancy, population size, and mean age of individuals living with each condition. We find that the observed mortality differences are primarily explained by differences in life expectancy, with the risk of death from deadly cancers outcompeting the risk of death from COVID-19.

## eLife assessment

This **valuable** work explores death coding data to understand the impact of COVID-19 on cancer mortality. The work provides **solid** evidence that deaths with cancer as a contributing cause were not above what would be expected during pandemic waves, suggesting that cancer did not strongly increase the risk of dying of COVID-19. These results are an interesting exploration into the coding of causes of death that can be used to make sense of how deaths are coded during a pandemic in the presence of other underlying diseases, such as cancer.

## Introduction

The dominant risk factors for COVID-19 mortality have consistently been shown to be advanced age, male gender, and certain chronic diseases such as diabetes, obesity, and heart disease (*Chavez-MacGregor et al., 2022*; *Rüthrich et al., 2021*; *Williamson et al., 2020*). Cancer has also been identified as a high-risk condition based on case-control and cohort studies, although these studies have provided conflicting results. In a large cohort study of ~500,000 COVID-19 inpatients, only cancer patients under recent treatment were at increased risk of COVID-19-related deaths (OR = 1.7) relative to non-cancer patients (*Chavez-MacGregor et al., 2022*). Conversely, a smaller European study of 3000 COVID-19 inpatients found that cancer was not a risk factor (*Rüthrich et al., 2021*), as did an international, multicenter study of 4000 confirmed COVID-19 inpatients (*Raad et al., 2023*).

**eLife digest** Establishing the true death toll of a pandemic like COVID-19 is difficult, as laboratory testing is generally too limited to directly count the number of deaths that can be attributed to a particular pathogen. To overcome this, researchers analyse excess mortality – that is, they compare the observed number of deaths with the expected level based on trends in prior years. These techniques have been used for over 100 years to estimate the burden of pandemic influenza and became a popular way to estimate deaths due to the COVID-19 pandemic.

Excess mortality can also reveal the impact of COVID-19 on sub-populations with chronic conditions. For example, previous studies showed that deaths with diabetes, heart disease and Alzheimer's disease listed as the primary cause of death increased during waves of COVID-19. Cancer deaths did not show such a pattern, however, despite some epidemiological studies identifying cancer as a risk factor for COVID-19 mortality.

To understand why this may be the case, Hansen et al. reviewed death certificates from different states in the United States during the first year of the pandemic. Their analyses of multiple-cause death records (listing cancer anywhere on the death certificate, not just as the primary cause of death) showed that death certificate coding practices during the pandemic did not explain the absence of excess cancer mortality. While a low level of excess mortality was detectable for cancers with longer life expectancy (breast cancer, for example), no elevation was observed for cancers with lower life expectancy, such as pancreatic cancer. The analyses demonstrate that the lack of excess mortality for especially deadly cancers can be explained through competing risks – in other words, the high risk of dying from the cancer itself vastly outweighs the additional risk posed by COVID-19.

These findings shed light on how competing mortality risks might mask the true impact of COVID-19 on cancer mortality and explain the apparent discrepancy between cohort studies and excess mortality studies. To fully comprehend the impact of COVID-19 on patients living with cancers, future research should look at the possibility of longer-term increases in cancer mortality due to late diagnosis during pandemic lockdowns, and an elevated risk of severe illness.

More recently a meta-analysis of 35 studies from Europe, North America, and Asia found a twofold increased risk of COVID-19 mortality among cancer patients (*Di Felice et al., 2022*). Similarly, a large analysis from the UK found that the risk of COVID-19 mortality for cancer patients had declined over the course of the pandemic but remained 2.5 times higher than for non-cancer patients into 2022 (*Starkey et al., 2023*). Taken together, such observational studies provide a mixed picture of cancer as a COVID-19 mortality risk factor, with several studies reporting that controlling for important factors such as age is a challenge. Furthermore, cancer is often considered as a single disease category despite the diversity of conditions and patients represented.

Further evidence for the relationship between cancer and COVID-19 comes from population-level analysis of vital statistics. A recent US study showed no elevation in underlying cancer deaths concomitant with COVID-19 waves, in stark contrast to the sharp rise in mortality from other chronic diseases (*Lee et al., 2023a*). In several other countries, including Sweden, Italy, Latvia, Brazil, England, and Wales, underlying cancer mortality was found to be stable or decreasing during the first year of the pandemic (*Alicandro et al., 2023*; *Fernandes et al., 2021*; *Gobina et al., 2022*; *Grande et al., 2022*; *Kontopantelis et al., 2022*; *Lundberg et al., 2023*). Further, an excess mortality study of 240,000 cancer patients in Belgium found a 33% rise in mortality in April 2020, but concluded that this was no different from the rise observed in the general population (*Silversmit et al., 2021*). The apparent lack of association between cancer mortality and COVID-19 on a population-level raises the question of the true relationship between cancer and COVID-19.

The relationship between these two diseases could occur via multiple biological mechanisms. First, immunosuppression in cancer patients could increase susceptibility to SARS-CoV-2 infection and/or risk of severe clinical outcome upon infection. Conversely, immunosuppression could act as a protective factor in the face of a severe respiratory infection that kills by over-stimulating the immune system – the immune incompetence rescue hypothesis (*Reichert et al., 2004*). This hypothesis was put forward to explain the observed absence in excess cancer mortality during the 1968 influenza pandemic, a departure from elevated mortality seen for other high-risk conditions such as heart

**Table 1.** Each diagnosis group and its corresponding ICD-10 codes, number of underlying deaths, mean age in years at time of death, the percentage of deaths occurring at home, and the percentage of deaths occurring in nursing homes for 2019 and 2020.

| Year | Diagnosis group | ICD-10 codes | No. deaths | Mean age, years (IQR) | %Home/ER | %Nursing home |
|------|-----------------|--------------|-----------|------------------------|----------|---------------|
|      | Cancer | C00-C99 | 493,397 | 72 (64–81) | 45 | 12 |
|      | Pancreatic cancer | C25 | 37,864 | 72 (64–80) | 51 | 9 |
|      | Colorectal cancer | C18-C20 | 42,484 | 71 (61–82) | 46 | 13 |
|      | Hematological cancers | C81-C96 | 47,174 | 74 (67–84) | 35 | 11 |
|      | Diabetes | E10-E14 | 70,763 | 72 (63–82) | 53 | 17 |
| 2019 | Alzheimer's | G30 | 98,675 | 87 (82–92) | 29 | 50 |
|      | Cancer | C00-C99 | 513,275 | 72 (64–81) | 55 | 8 |
|      | Pancreatic cancer | C25 | 39,893 | 72 (65–80) | 61 | 6 |
|      | Colorectal cancer | C18-C20 | 43,990 | 71 (61–82) | 56 | 9 |
|      | Hematological cancers | C81-C96 | 49,161 | 74 (67–84) | 46 | 8 |
|      | Diabetes | E10-E14 | 88,124 | 71 (62–82) | 58 | 15 |
| 2020 | Alzheimer's | G30 | 115,256 | 86 (82–92) | 33 | 46 |

disease and diabetes (*Reichert et al., 2004*). A further mechanism that could affect the observed relationship between cancer deaths and COVID-19 is changing guidelines for establishing the primary cause of death. Coding guidelines evolved throughout the pandemic as testing for SARS-CoV-2 infection became more widespread, which presumably affected vital statistics studies.

To further elucidate the relationship between cancer mortality and COVID-19 on a population level, we analyzed US vital statistics in detail to understand the potential role of death certificate coding changes during the pandemic and explored putative differences in mortality patterns between different types of cancer. We considered death certificates listing cancer as the underlying cause (UC) of death or anywhere on the death certificate (multiple cause [MC]). Assuming there is a high propensity to attribute a primary code of COVID-19 during the pandemic in any patient with COVID-19, deaths among individuals with both cancer and COVID-19 near the time of death would be coded as UC COVID-19. However, cancer should still be captured in the MC data, and thus, analysis of MC death data should control for any changes in death certificate coding practices during the pandemic (*Fedeli et al., 2024*). The US provides a particularly useful case study as the timing of COVID-19 waves varied considerably between states, so that elevations in cancer deaths, should they exist, should also be heterogeneous. For comparison, we also assessed population-level excess mortality patterns for other chronic conditions such as diabetes, ischemic heart disease (IHD), kidney disease, and Alzheimer's, for which the association with COVID-19 is less debated.

## Results

### Establishing patterns and timing of COVID-19-related deaths

We obtained individual ICD-10-coded death certificate data from the US for the period January 1, 2014, to December 31, 2020. We compiled time series by week and cause of death, for UC and for MC mortality. We considered 10 causes of death, including diabetes, Alzheimer's disease, IHD, kidney disease, and six types of cancer (all-cause cancer, colorectal, breast, pancreatic, lung, and hematological; see *Table 1* and *Appendix 1—table 1* for a list of disease codes). We chose these specific cancers to illustrate conditions for which the 5-year survival rate is low (13% and 25%, respectively, for pancreatic and lung cancers) and high (65% and 91%, respectively, for colorectal and breast cancers) (*National Cancer Institute, 2024*). Hematological cancer (67% 5-year survival rate) was included because it has been singled out as a risk factor in several previous studies due to the immune

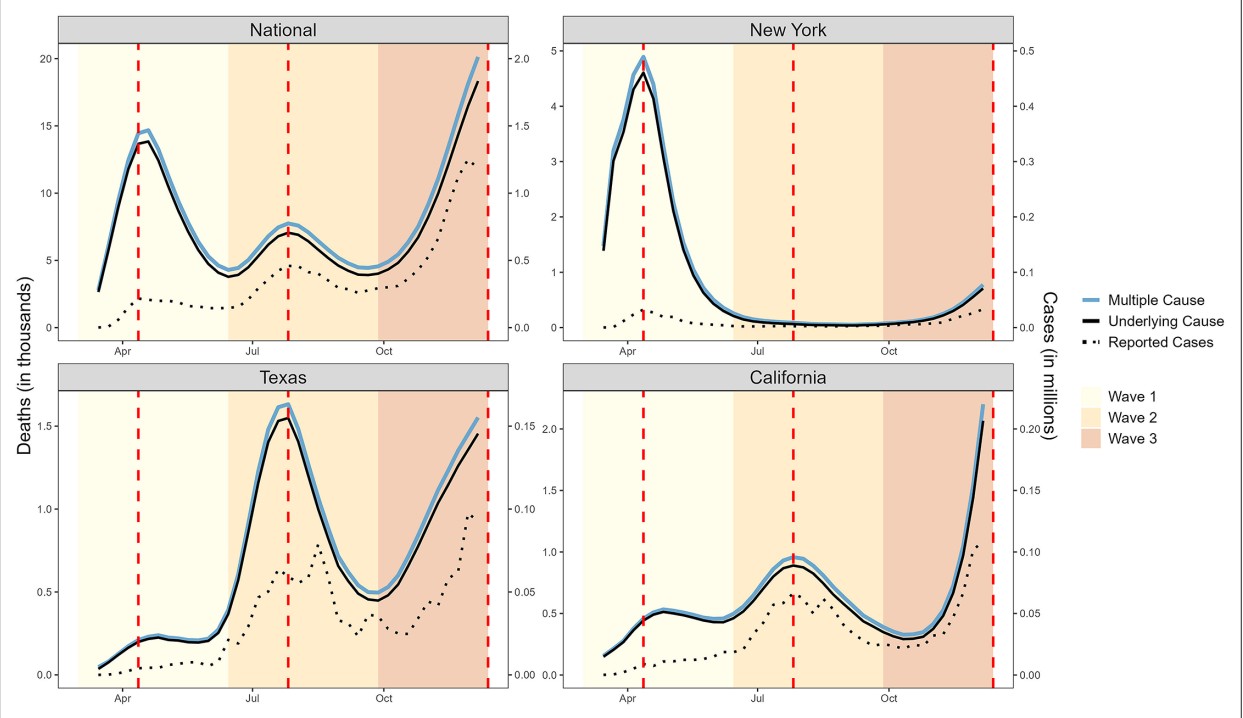

**Figure 1.** Weekly counts of death certificates listing COVID-19 as either the underlying or a multiple cause. When included on a death certificate, COVID-19 was most often listed as the underlying cause of death rather than a contributing cause. National-level data reveal three distinct waves: Wave 1 (spring, March 1 to June 27, 2020), Wave 2 (summer, June 28 to October 3, 2020), and Wave 3 (winter, October 4 to December 6, 2020, incomplete). Vertical dashed lines represent the peak of each wave, dotted lines represent the number of reported cases (y-axis on the right). New York experienced its first large COVID-19 wave in Wave 1, while Texas had its first large wave in Wave 2 and California did not experience a large wave until Wave 3 which had not yet peaked at the end of 2020.

suppression associated with both its malignancy and treatment (***Chavez-MacGregor et al., 2022***; ***Han et al., 2022a***; ***Rüthrich et al., 2021***; ***Williamson et al., 2020***). To compare mortality patterns with the timing of COVID-19 pandemic waves, we accessed national- and state-level counts of reported COVID-19 cases from the ***Centers for Disease Control and Prevention, 2022***.

In national data, time series of COVID-19-coded death certificates (both UC and MC) tracked with the temporal patterns of laboratory-confirmed COVID-19 cases (***Figure 1***), revealing three distinct COVID-19 waves: a spring wave peaking on April 12, 2020, a smaller summer wave peaking on July 26, 2020, and a large winter wave that had not yet peaked by the end of the study in December 2020. This correspondence between COVID-19 case and death activity represents a 'signature' mortality pattern of COVID-19.

Analysis of state-level data reveals variable timing, intensity, and number of COVID-19 waves across the US during 2020. To focus on periods with substantial COVID-19 activity and explore the association with cancer, we identified three large US states with unique, well-defined waves (***Figure 1***). New York (NY) state experienced a large, early wave in March–May 2020, based on recorded COVID-19 cases and deaths and high seroprevalence of SARS-CoV-2 antibodies in this period (over 20%; ***Stadlbauer et al., 2021***). Meanwhile, California (CA) experienced a large COVID-19 wave at the end of the year and had little activity during the spring and summer. Finally, Texas (TX) had two large waves; one during late summer, followed by one in winter 2020.

## National patterns in excess mortality from cancer

Similar to other influenza and COVID-19 population-level mortality studies (***Islam et al., 2021***; ***Karlinsky and Kobak, 2021***; ***Lee et al., 2023a***; ***Msemburi et al., 2023***), we established a weekly baseline model for expected mortality in the absence of pandemic activity by modeling time trends and seasonality in pre-pandemic data and letting the model run forward during the pandemic (see Materials and methods). Each cause of death (UC and MC) and geography (aggregated National, NY,

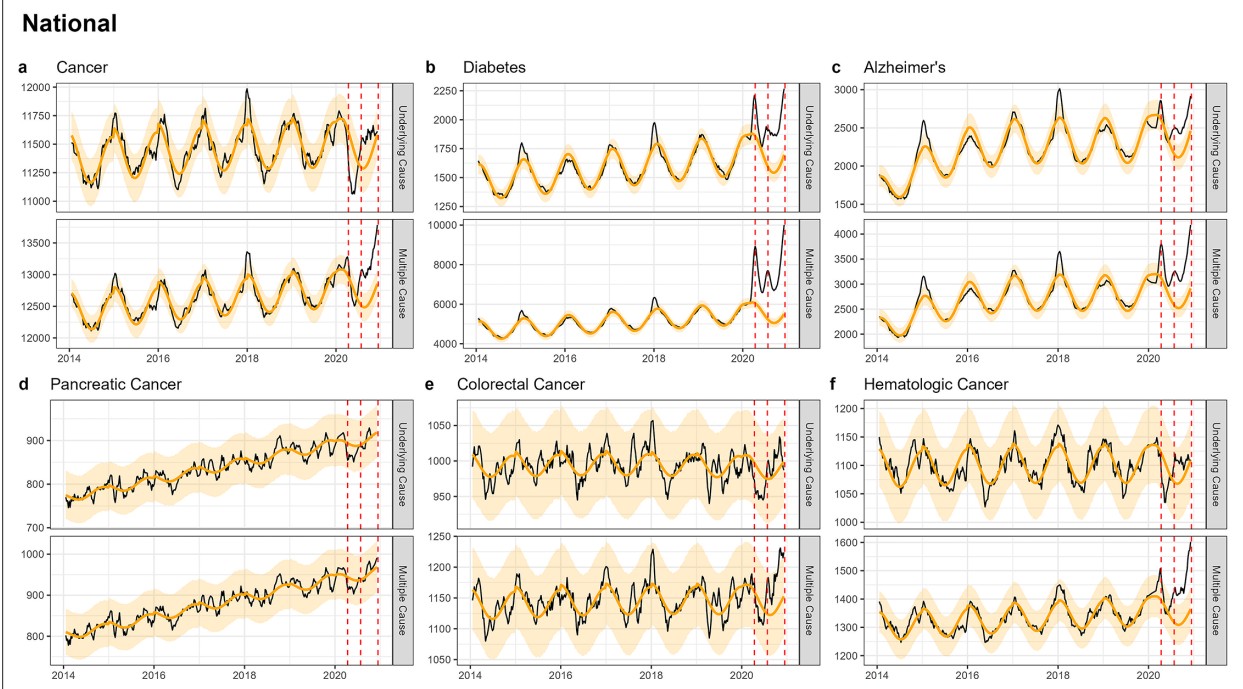

**Figure 2.** National-level weekly observed and estimated baseline mortality for each diagnosis group (Cancer (**a**), Diabetes (**b**), Alzheimer's (**c**), Pancreatic Cancer (**d**), Colorectal Cancer (**e**), Hematologic Cancer (**f**)) as both the underlying cause or anywhere on the death certificate (multiple cause) from 2014 to 2020. Red dashed lines represent the timing of the peaks for the three COVID-19 waves in 2020. Baselines during the pandemic are projected based on the previous years of data.

TX, and CA) was modeled separately. We then computed excess mortality as the difference between observed deaths and the model-predicted baseline. We summed weekly estimates to calculate excess mortality for the full pandemic period and during each of the three waves (see Materials and methods). In addition to these absolute effects of the pandemic on mortality, we also calculated the relative effects by dividing excess mortality by baseline mortality. This approach has been used in the past to standardize mortality effects in strata with very different underlying risks (e.g. age groups, geographies, or causes of death, see Materials and methods).

Nationally, we found a drop in UC cancer deaths during spring 2020 (*Figure 2a*; *Table 2*), although the drop was not statistically significant. A similar non-significant decline was also seen for specific cancer types (*Figure 2d–f*; *Appendix 1—figure 1a, f–j*). Further, pre-pandemic mortality trends for each cancer type continued unabated during the first pandemic year. We reasoned that this early drop in UC cancer deaths may be explained by changes in coding practices, so we next turned to MC mortality to resolve this question.

Time series of MC cancer mortality showed a significant increase in all three waves (*Figure 2a*; *Appendix 1—table 2*). A similar pattern was seen in MC time series for colorectal (*Figure 2h*), breast (*Appendix 1—figure 1i*), and hematological cancer (*Appendix 1—figure 1j*). However, the total excess mortality was modest with 13,600 excess cancer deaths in 2020, representing a statistically significant 3% elevation over baseline (*Table 2*). The largest relative increase in MC mortality was observed in hematological cancer at 7% (statistically significant, 3600 excess deaths). No excess in MC mortality was seen for the two deadliest cancers, pancreatic cancer (*Figure 2f*) and lung cancer (*Appendix 1—figure 1g*).

## National patterns in deaths due to other chronic conditions

We considered diabetes and Alzheimer's as 'positive controls' as they are also considered COVID-19 risk factors and can illustrate how positive associations between chronic conditions and COVID-19 manifest in population-level excess mortality studies. Diabetes provides a particularly useful comparator for cancer as the mean age-at-death is similar (~72 years, *Table 1*) and because few individuals live in a nursing home (Appendix 1 - Supplemental Methods). Mortality time series from UC and MC

**Table 2.** The estimated number of excess deaths and the percentage over baseline for each diagnosis group when listed as both the underlying cause or anywhere on the death certificate (multiple cause).

Estimates for the national-level data are provided for the full pandemic period and for each state based on when the first large wave was experienced.

| Cause of death | State | Wave | Multiple cause | | Underlying cause | |
|---|---|---|---|---|---|---|
| | | | Excess deaths | % Over baseline | Excess deaths | % Over baseline |
| | National | Overall | 13601* | 3.0 | 11 | 0.0 |
| | New York | 1 | 747 | 6.0 | –474 | –5.0 |
| | Texas | 2 | 467 | 4.0 | 39 | 0.0 |
| Cancer | California | 3 | 529 | 4.0 | 82 | 1.0 |
| | National | Overall | –25 | –0.0 | –282 | –1.0 |
| | New York | 1 | 8 | 1.0 | –16 | –2.0 |
| | Texas | 2 | 17 | 2.0 | 24 | 3.0 |
| Pancreatic cancer | California | 3 | 0 | 0.0 | –18 | –2.0 |
| | National | Overall | 988 | 2.0 | –168 | –0.0 |
| | New York | 1 | 91 | 9.0 | –16 | –2.0 |
| | Texas | 2 | 4 | 0.0 | –34 | –3.0 |
| Colorectal cancer | California | 3 | 27 | 2.0 | –1 | –0.0 |
| Hematological cancers | National | Overall | 3615* | 7.0 | 111 | 0.0 |
| | New York | 1 | 121 | 10.0 | –107 | –11.0 |
| | Texas | 2 | 136 | 11.0 | 21 | 2.0 |
| | California | 3 | 114 | 8.0 | 20 | 2.0 |
| | National | Overall | 82,318* | 37.0 | 10,784* | 16.0 |
| | New York | 1 | 5945* | 128.0 | 568* | 40.0 |
| | Texas | 2 | 4612* | 77.0 | 420* | 23.0 |
| Diabetes | California | 3 | 3474* | 59.0 | 575* | 33.0 |
| | National | Overall | 21,712* | 19.0 | 8528* | 9.0 |
| | New York | 1 | 734* | 49.0 | 188 | 16.0 |
| | Texas | 2 | 1398* | 45.0 | 805* | 31.0 |
| Alzheimer's | California | 3 | 726* | 18.0 | 259 | 8.0 |

*Confidence interval does not include zero.

diabetes and Alzheimer's were highly correlated with COVID-19 activity, with statistically significant mortality elevation synchronous with pandemic waves (*Figure 2b and c*; *Appendix 1—figures 2–5*). For diabetes, we measured an excess of 10,800 and 82,300 deaths (UC and MC, respectively), corresponding to statistically significant elevations of 16% and 37% over baseline level mortality (*Table 2*). For Alzheimer's, we estimated 8500 and 21,700 excess deaths, corresponding to statistically significant elevations of 9% and 19% elevation over baseline, respectively. Pandemic-related excess mortality was also seen for IHD and kidney disease (see supplement for estimates, *Appendix 1—table 2*).

## State-level patterns in excess mortality

Similar to patterns seen in national-level data, none of the state-level analyses revealed notable increases in UC cancer mortality, while there was a modest, non-significant increase in MC cancer

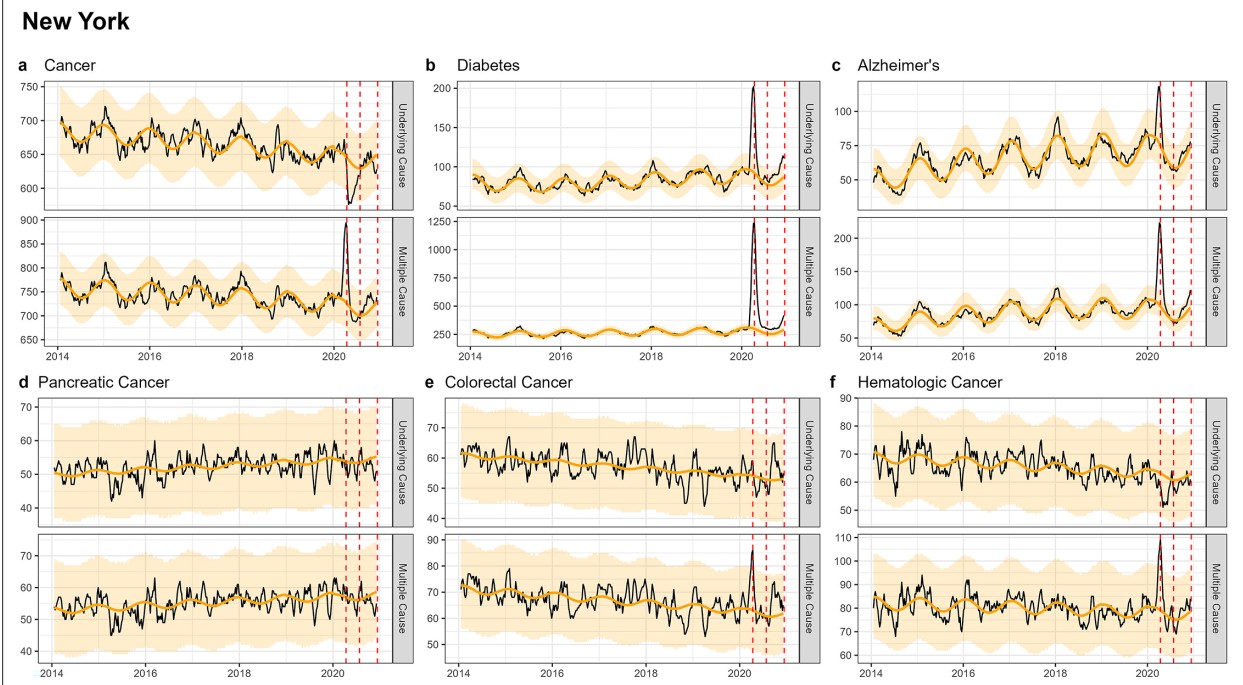

**Figure 3.** Weekly observed and estimated baseline mortality for each diagnosis group (Cancer (**a**), Diabetes (**b**), Alzheimer's (**c**), Pancreatic Cancer (**d**), Colorectal Cancer (**e**), Hematologic Cancer (**f**)) as both the underlying cause or anywhere on the death certificate (multiple cause) from 2014 to 2020 in New York. Red dashed lines represent the timing of the peaks for the three COVID-19 waves in 2020. New York experienced its first large wave of COVID-19 in spring 2020 (Wave 1).

mortality (*Figures 3–5*; *Appendix 1—figures 6–8*). The largest mortality increase was seen in NY during the spring wave, with a 6% rise in MC cancer mortality above the model baseline (*Table 2*; *Appendix 1—table 3*). The magnitude of the increase seen during the spring wave varied by cancer type, with minimal increases seen in pancreatic and lung cancers (1%) and higher increases in colorectal, hematological, and breast cancers (9%, 10%, and 16%, respectively). For comparison, there was a statistically significant rise in Alzheimer's and diabetes deaths during this wave of 49% and 128%.

In CA and TX, mortality fluctuations were less pronounced than in NY, coinciding with less intense COVID-19 waves, and this was seen across all conditions. MC excess mortality estimates remained within ±4% of baseline levels for cancers, irrespective of the type of cancer and pandemic wave, except for hematological cancers which saw an 11% rise in TX during the summer wave and an 8% rise in CA during the winter wave. None of these elevations were statistically significant. In comparison, there was statistically significant excess mortality elevation for both Alzheimer's and diabetes deaths (range, 18–59% in the CA winter wave, and 45–77% in the TX summer wave, *Table 2*, *Appendix 1— tables 4 and 5*).

## Demographic mortality projections under the null hypothesis that cancer in and of itself is not a risk factor for COVID-19 mortality

Next, to get a sense of the expected mortality elevation, we ran simulations to gauge the level of individual-level association (traditionally measured as relative risk [RR]) between COVID-19 and the studied chronic conditions that is consistent with the population-level excess mortality patterns observed during the pandemic. Using cancer as an example, two main factors could drive cancer mortality patterns during COVID-19, namely the size and age of the population living with cancer (since age is such a pronounced risk factor for COVID-19), and the life expectancy under cancer diagnosis. These factors would operate irrespective of the true biological relationship between COVID-19 severity and cancer. The same logic applies to mortality from other chronic conditions, such as diabetes or Alzheimer's.

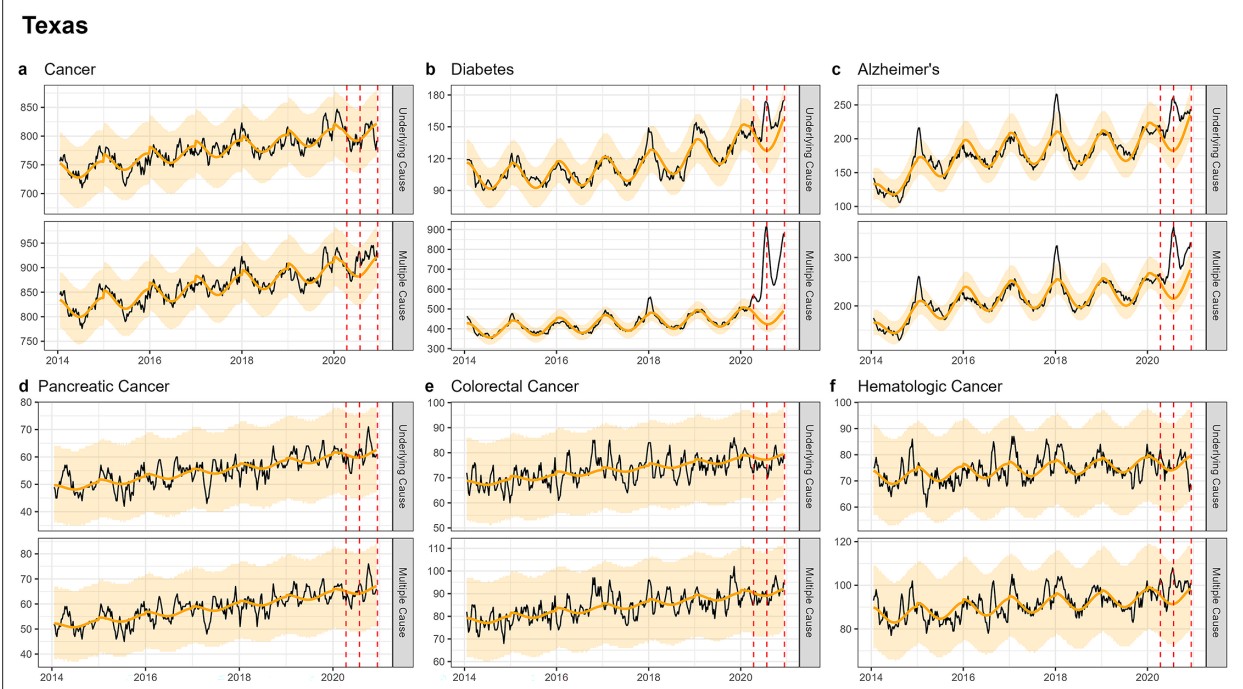

**Figure 4.** Weekly observed and estimated baseline mortality for each diagnosis group (Cancer (**a**), Diabetes (**b**), Alzheimer's (**c**), Pancreatic Cancer (**d**), Colorectal Cancer (**e**), Hematologic Cancer (**f**)) as both the underlying cause or anywhere on the death certificate (multiple cause) from 2014 to 2020 in Texas. Red dashed lines represent the timing of the peaks for the three COVID-19 waves in 2020. Texas experienced its first large wave of COVID-19 in the summer of 2020 (Wave 2).

To test the hypothesis that these population factors alone could explain differences in excess mortality between chronic conditions, we designed a simple model of COVID-19 mortality for individuals with chronic conditions (see Materials and methods for details). The model projected excess mortality during the pandemic under the null hypothesis that the chronic condition was not in and of itself a risk factor for COVID-19 mortality, with only the demography of the population living with the disease (namely, the age and size of the at-risk populations and baseline risk of death from each condition) affecting excess mortality. In the demographic model, we first estimated the number of expected SARS-CoV-2 infections among persons with a certain condition, by multiplying the estimated number of US individuals living with the condition (***CDC, Division of Population Health, 2022***; ***Dhana et al., 2023***; ***Rajan et al., 2021***; ***U.S. Cancer Statistics Working Group, 2023***) by the reported SARS-CoV-2 seroprevalence at the end of our study period (December 2020 for the national, or after each wave for the state data) (***Centers for Disease Control and Prevention, 2023***). We focused on seroprevalence among individuals ≥65 years, the most relevant age group for the conditions we considered (we also run a sensitivity analysis considering seroprevalence in adults 50–64 years, see Discussion). We then multiplied the estimated number of SARS-CoV-2 infections by an age-specific infection-fatality ratio (IFR) for SARS-CoV-2 (***COVID-19 Forecasting Team, 2022***). This gave an estimate of COVID-19-related deaths, or excess deaths, for a given condition. To estimate a percent elevation over baseline and compare with our vital statistics analysis, we divided the excess death estimate derived from the demographic model by the total deaths for that condition for a similar period of time in 2019 (see Materials and methods). We repeated this analysis for each cancer type, diabetes, and Alzheimer's. In addition to the null hypothesis, we also projected alternative hypotheses of a biological association between chronic conditions and COVID-19, assuming that a given chronic condition would raise the risk of COVID-19 mortality (via the IFR) by a factor of 2 or 5. We compared these modeled expectations for the null and alternative hypotheses with the observed excess mortality in 2020, using MC mortality as the outcome (***Table 2***).

Under the null hypothesis we projected a 0–2% elevation over the 2019 baseline in deaths for all cancer types in national data, and 0–9% elevations in state-level data (***Table 3***). Under the alternative hypothesis that cancer increases COVID-19 mortality risk by a factor of 2, the projected elevation

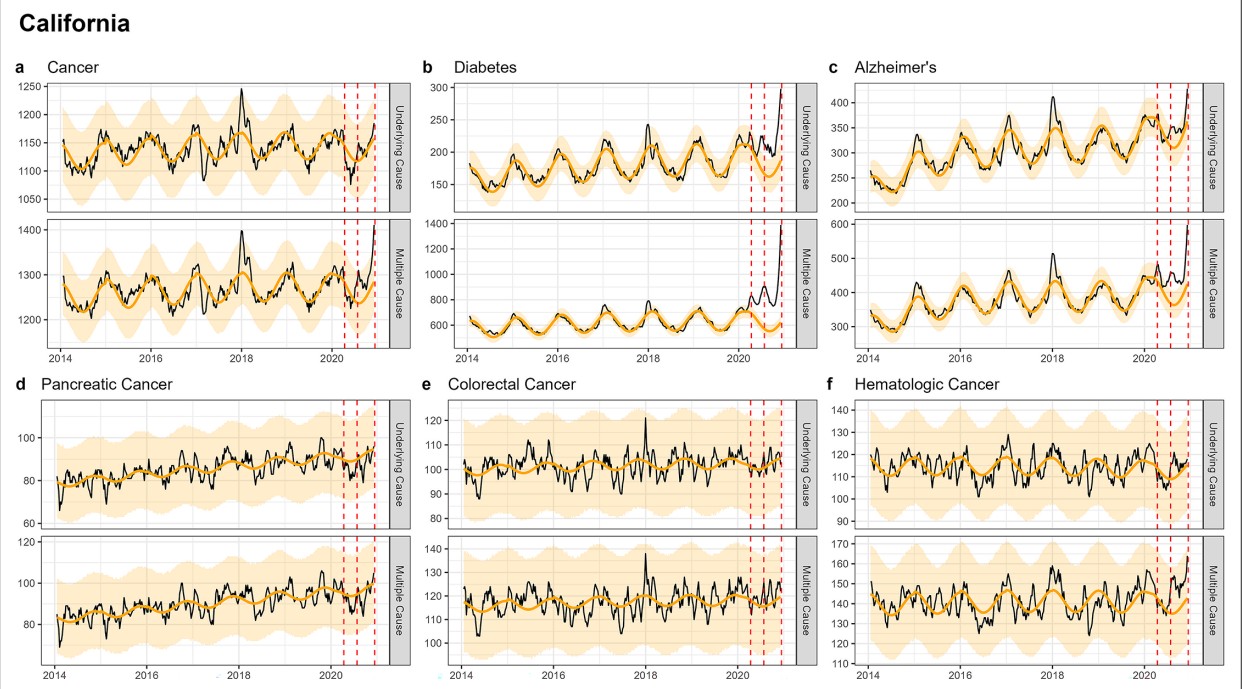

**Figure 5.** Weekly observed and estimated baseline mortality for each diagnosis group (Cancer (**a**), Diabetes (**b**), Alzheimer's (**c**), Pancreatic Cancer (**d**), Colorectal Cancer (**e**), Hematologic Cancer (**f**)) as both the underlying cause or anywhere on the death certificate (multiple cause) from 2014 to 2020 in California. Red dashed lines represent the timing of the peaks for the three COVID-19 waves in 2020. California did not experience a large wave of COVID-19 until the winter of 2020–2021 (Wave 3), only the first half of which is captured here.

is 0–5% in national data and 0–18% in state-level data. In general, the largest projected increases were found in NY state, driven by the higher attack rates. We also see systematic differences in the percent elevation over baseline by type of cancer, related to the lethality of different cancers. For instance, even if cancer increases COVID-19 mortality risk by a factor of 2, we expect to see only a 0–1% increase for particularly deadly cancers such as pancreatic and lung cancer, in part driven by the high competing risk of death from these cancers (short life expectancy) and the small size of the population-at-risk. The expected increases for less deadly cancers, such as colorectal and breast, were notably higher (2–5% in national data, and 9–18% during the large spring wave in NY), in part driven by the lower risk of death from these cancers (longer life expectancy). Based on the observations from our time series analysis of MC mortality in all states, non-hematological cancers are most consistent with a one- to twofold increase in mortality, with the caveat that most of the confidence intervals include zero, and the differences in projected mortality under these hypotheses are minimal. In contrast, for hematological cancer, the observed rise in mortality exceeds the expected elevation even under the assumption of a fivefold increase in mortality.

We repeated this analysis for diabetes and Alzheimer's (*Table 3*). For diabetes under the null hypothesis, we projected an 8% elevation over baseline in national data and 12–30% in state-level data based on the age distribution and substantial size of the population-at-risk alone. In fact, we observed in vital statistics analysis a 37% elevation over baseline in national US data and 59–128% in state-level data, with the largest increase seen in NY and lowest increase in CA. These observations are most consistent with a fivefold increase in mortality based on our demographic model (projected elevation 40% nationally and 62–148% at the state level). For Alzheimer's under the null hypothesis, we projected a 28% increase over baseline nationally, and 30–191% increases at the state level, largely driven by the advanced age of the population-at-risk. In contrast, analysis of vital statistics data reveals a 19% increase nationally and 18–49% across states, which is in fact lower than the null hypothesis (we return to this surprising result in the Discussion). Strikingly, our demographic model supports that COVID-19 will manifest differently in population-level excess mortality for each of these chronic conditions, even under the null hypothesis of no biological association between viral infection and these underlying comorbidities. Overall, these projections support the idea that demography alone

**Table 3.** Projections of COVID-19-related excess mortality patterns for different cancers and chronic conditions in the US, under different hypotheses for the association between the condition and COVID-19.

Projections are provided for the null hypothesis of no biological interaction between the condition and COVID-19; these projections are solely driven by the size and mean age of the population living with each condition (where age determines the infection-fatality ratio from COVID-19), and the baseline risk of death from the condition over a similar time period (March to December 2019 for the national data, and for the states comparable dates in 2019 corresponding to the relevant COVID-19 wave). Additional projections are provided under alternative hypotheses, where each condition is associated with a relative risk (RR) of 2 or 5 for COVID-19-related death (infection-fatality ratio multiplied by 2 or 5).

| Chronic condition | State | Population-at-risk | Mean age | Wave | Observed MC deaths over same period in 2019 | Observed excess deaths over (% over baseline) in 2020 | Expected excess (null) | Expected excess (RR = 2) | Expected excess (RR = 5) |
|---|---|---|---|---|---|---|---|---|---|
| All cancers | National | 5,718,925 | 65 | Overall | 546,453 | 3 (1–4) | 1 (1–2) | 2 (1–4) | 6 (4–10) |
| | New York | 400,891 | 65 | Wave 1 | 12,244 | 6 (–1 to 15) | 4 (2–10) | 9 (3–20) | 22 (8–51) |
| | Texas | 397,993 | 63 | Wave 2 | 12,187 | 4 (–3 to 11) | 2 (1–6) | 5 (2–12) | 11 (4–29) |
| | California | 599,552 | 64 | Wave 3 | 16,713 | 4 (–1 to 10) | 2 (0–5) | 4 (1–9) | 9 (2–23) |
| Pancreatic | National | 66,319 | 67 | Overall | 39,798 | 0 (–6 to 7) | 0 (0–0) | 0 (0–1) | 1 (1–2) |
| | New York | 2584 | 67 | Wave 1 | 963 | 1 (–21 to 35) | 0 (0–1) | 1 (0–2) | 2 (1–5) |
| | Texas | 2264 | 66 | Wave 2 | 882 | 2 (–19 to 34) | 0 (0–1) | 0 (0–1) | 1 (0–3) |
| | California | 3482 | 67 | Wave 3 | 1277 | 0 (–17 to 24) | 0 (0–0) | 0 (0–1) | 1 (0–2) |
| Lung cancer | National | 425,015 | 70 | Overall | 123,622 | 1 (–3 to 5) | 1 (0–1) | 1 (1–2) | 3 (2–5) |
| | New York | 17,709 | 71 | Wave 1 | 2643 | 1 (–13 to 20) | 2 (1–4) | 3 (1–8) | 8 (3–20) |
| | Texas | 12,700 | 70 | Wave 2 | 2513 | 2 (–11 to 20) | 1 (0–2) | 1 (1–4) | 4 (1–9) |
| | California | 19,079 | 70 | Wave 3 | 2861 | 3 (–10 to 18) | 1 (0–2) | 1 (0–3) | 3 (1–8) |
| Hematological | National | 459,463 | 62 | Overall | 57,892 | 7 (1–13) | 1 (0–1) | 1 (1–2) | 3 (2–5) |
| | New York | 15,577 | 62 | Wave 1 | 1305 | 10 (–11 to 40) | 1 (0–3) | 2 (1–5) | 6 (2–13) |
| | Texas | 14,927 | 59 | Wave 2 | 1231 | 11 (–9 to 38) | 1 (0–1) | 1 (0–3) | 3 (1–7) |
| | California | 21,290 | 61 | Wave 3 | 1916 | 8 (–8 to 29) | 0 (0–1) | 1 (0–2) | 2 (1–5) |

*Table 3 continued on next page*

Table 3 continued

| Chronic condition | State | Population-at-risk | Mean age | Wave | Observed MC deaths over same period in 2019 | Observed excess (% over baseline) in 2020 | Expected excess (null) | Expected excess (RR = 2) | Expected excess (RR = 5) |
|---|---|---|---|---|---|---|---|---|---|
| Colorectal | National | 473,264 | 66 | Overall | 49,053 | 2 (−4 to 8) | 1 (1–2) | 2 (1–4) | 6 (4–10) |
| | New York | 30,859 | 66 | Wave 1 | 1048 | 9 (−13 to 44) | 4 (2–10) | 9 (3–20) | 22 (8–51) |
| | Texas | 36,641 | 65 | Wave 2 | 1224 | 0 (−18 to 26) | 3 (1–7) | 5 (2–13) | 13 (4–33) |
| | California | 51,863 | 65 | Wave 3 | 1575 | 2 (−14 to 24) | 2 (1–5) | 4 (1–10) | 9 (3–24) |
| Breast | National | 1,097,917 | 64 | Overall | 43,519 | 2 (−4 to 9) | 2 (2–4) | 5 (3–8) | 12 (8–21) |
| | New York | 74,459 | 64 | Wave 1 | 981 | 16 (−8 to 53) | 9 (3–21) | 18 (7–42) | 46 (17–106) |
| | Texas | 77,860 | 62 | Wave 2 | 1019 | 3 (−17 to 32) | 5 (2–12) | 10 (3–24) | 24 (8–61) |
| | California | 123,433 | 63 | Wave 3 | 1421 | 2 (−15 to 25) | 4 (1–10) | 8 (2–20) | 20 (5–51) |
| Diabetes | National | 29,105,146 | 60 | Overall | 229,326 | 37 (31–43) | 8 (5–14) | 16 (10–28) | 40 (26–69) |
| | New York | 1,792,926 | 60 | Wave 1 | 4804 | 128 (104–158) | 30 (11–68) | 59 (22–136) | 148 (55–340) |
| | Texas | 2,450,005 | 58 | Wave 2 | 5898 | 77 (61–96) | 17 (6–44) | 35 (12–87) | 86 (30–218) |
| | California | 3514,440 | 59 | Wave 3 | 8399 | 59 (47–74) | 12 (3–32) | 25 (7–64) | 62 (17–160) |
| Alzheimer's | National | 6,070,000 | 81 | Overall | 118,993 | 19 (11–28) | 28 (18–48) | 57 (36–96) | 142 (90–240) |
| | New York | 426,500 | 81 | Wave 1 | 1563 | 49 (23–87) | 191 (70–432) | 381 (140–863) | 953 (350–2158) |
| | Texas | 459,300 | 80 | Wave 2 | 2974 | 45 (27–69) | 63 (21–158) | 126 (43–315) | 315 (107–788) |
| | California | 719,700 | 81 | Wave 3 | 5394 | 18 (6–33) | 39 (11–98) | 78 (21–196) | 195 (53–491) |

(age, size, and baseline mortality of the population living with each of these conditions) can explain much of the differences in absolute and relative mortality elevations seen during the pandemic across conditions like cancer, diabetes, and Alzheimer's.

## Discussion

Cancer is generally thought of as a risk factor for severe COVID-19 outcomes, yet observational studies have produced conflicting evidence. With recent availability of more detailed US vital statistics data, we used statistical time series approaches to generate excess mortality estimates for MC death data, different types of cancer, and several geographical locations during 2020. We accounted for potential changes in coding practices during the pandemic, for instance capturing a COVID-19 patient with cancer whose death may have been coded as an underlying COVID-19 death and not a cancer death. Based on MC death data, we estimated 13,600 national COVID-19-related excess cancer deaths, which aligns well with reporting on death certificate data, where 13,400 deaths are ascribed to COVID-19 in cancer patients (*Appendix 1—figure 9*; *Fedeli et al., 2024*). Yet these deaths only represent a 3% elevation over the expected baseline cancer mortality. Percent mortality elevation was measurably higher for less deadly cancers (breast and colorectal) than cancers with a poor 5-year survival (lung and pancreatic). Consistent with other studies (*Chavez-MacGregor et al., 2022*; *Han et al., 2022b*; *Rüthrich et al., 2021*; *Williamson et al., 2020*), we found that the largest mortality increase for specific cancer types was seen in hematological cancers with a 7% elevation over baseline in national data. Across the board, the largest elevations in cancer mortality were observed

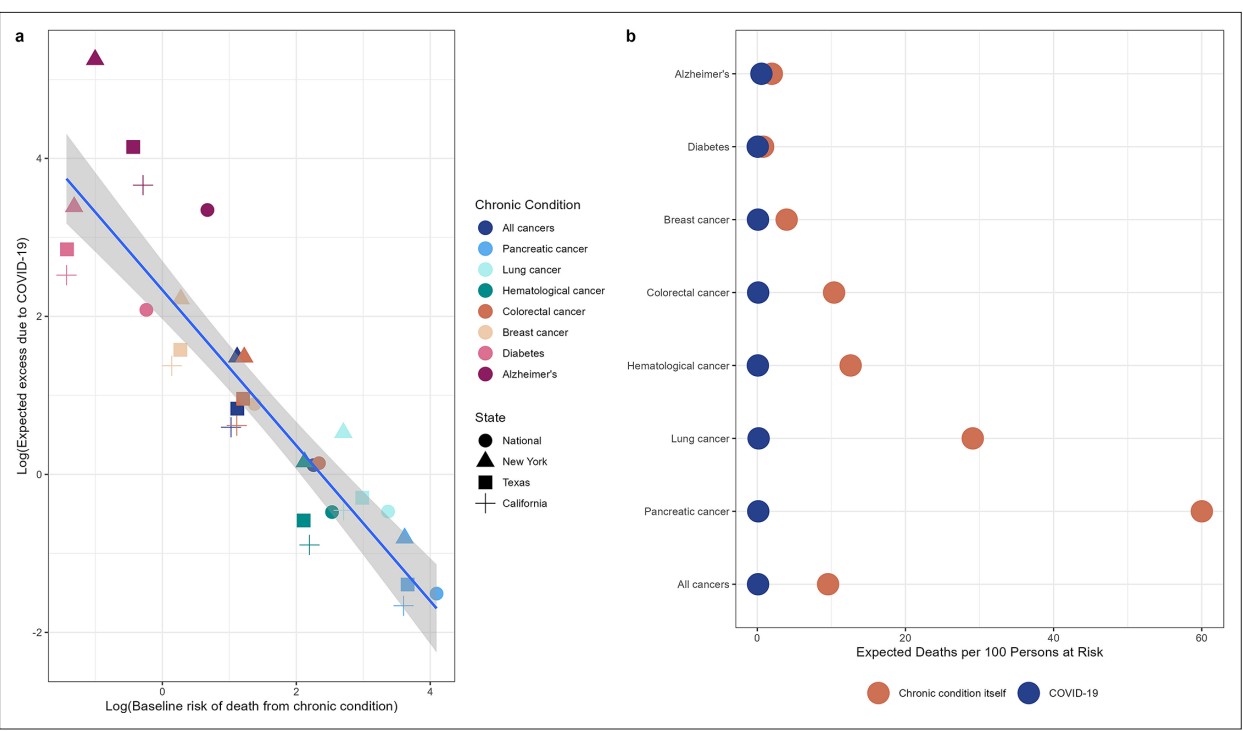

**Figure 6.** Illustration of competing risks. Based on our demographic model, we expect a small increase in cancer mortality relative to diabetes and Alzheimer's due to the higher competing risk of death from cancer compared to COVID-19. Panel (a) shows the log of the baseline mortality rate (based on observed mortality in 2019) from each condition on the x-axis and the log of the expected excess mortality (elevation over baseline) on the y-axis. Chronic conditions are shown in colors while states are shown in different shapes. Pancreatic cancer, the deadliest cancer considered, is on the bottom right (highest baseline mortality, lowest expected excess) while diabetes and Alzheimer's are on the top left (lowest baseline mortality, highest expected excess). Panel (b) shows the baseline number of deaths per 100 persons at risk for each condition expected from March to December 2020 (based on deaths over this same period in 2019, orange dots) compared to the expected number of deaths due to COVID-19 under the null hypothesis (blue dots). The null hypothesis stipulates that there is no biological association between any of these chronic diseases and COVID-19. For diabetes and Alzheimer's, the baseline risks of death are similar to the risk of death from COVID-19, while risk of death from cancer outcompetes risk of COVID-19 death for all types of cancer.

in the states most impacted by the first year of the COVID-19 pandemic (e.g. NY), lending support to the specificity of our excess mortality approach.

In contrast to cancer, we observed substantial COVID-19-related excess mortality for diabetes and Alzheimer's, temporally and geographically consistent with the three-wave 'signature' pattern observed in reported COVID-19 cases and deaths across the US. To investigate whether demographic differences in underlying patient populations (age distribution, population size, and baseline risk of death due to chronic condition) could explain differences in excess mortality during the pandemic, we ran a simple demographic model for each condition – first assuming the condition in and of itself was not a risk factor for COVID-19-related mortality (null hypothesis). Doing so we found that the rise in cancer deaths during COVID-19 was expected to remain low compared to these other chronic conditions, largely driven by the higher risk of death from cancer itself compared to diabetes and Alzheimer's. These demographic projections illustrate the importance of competing risks (*Figure 6*), where the risk of cancer death predominates over the risk of COVID-19 death in 2020. This is exacerbated in cancers with high mortality rates. For instance, even if pancreatic cancer had in fact doubled the risk of dying of COVID-19 (IFR = 4.2% vs 2.1%), we would only expect a rise in excess mortality around 0.4% during the pandemic (*Table 3*), while the 2019 baseline risk of death for pancreatic cancer itself is over 60% (*Figure 6*). On the other hand, for conditions with a lower baseline level mortality, such as diabetes, we expect substantial COVID-19-driven elevations in mortality.

Our analysis revealed interesting differences between types of cancers. Both nationally and at the state level, the observed excess mortality for non-hematological cancers was consistent with a one- to twofold increase in COVID-19 mortality risk in patients with these types of cancer. Importantly, our analysis ignores any behavioral effects associated with the pandemic. It is conceivable that cancer patients may have shielded themselves from COVID-19 more than the average person in 2020. Our projections assume an average risk of infection for a typical individual over 65 years as there is no serologic data on infection attack rates for specific clinical population subgroups (of any age). If shielding from exposure to SARS-CoV-2 was high among cancer patients, our projections of cancer excess mortality during the pandemic would be inflated. In other words, if shielding was particularly pronounced, cancer may conceivably be a higher risk factor than shown here. Retrospective serologic analysis of banked sera from the first year of the pandemic, broken down by underlying comorbidities, may shed light on whether infection risk may have varied by chronic condition.

State-level mortality patterns can potentially provide complementary insights on the question of shielding. Because NY state experienced the earliest and most intense COVID-19 wave of the US, with over 20% of the population infected in spring 2020 (*Stadlbauer et al., 2021*), and because social distancing did not come into effect until March 2020, shielding would have had a more limited impact there than in other states. Thus, a biological relationship between cancer and COVID-19 would have been most dramatic in NY in spring 2020. Indeed, cancer excess mortality was exacerbated in NY, including an 9–16% increase in colorectal and breast cancer mortality, consistent with a twofold increase in COVID-19 mortality risk from these cancers, and a 10% increase in hematological cancers, consistent with a fivefold increase in COVID-19 mortality risk. In NY, the absence of excess mortality in lethal cancers, such as pancreatic and lung cancers (1% over baseline) are, as discussed above, still consistent with what would be expected under a high competing risk situation.

We used diabetes and Alzheimer's as positive controls for a known biological association between COVID-19 and chronic conditions. Diabetes stood out in our analyses with the highest absolute and relative increases in excess mortality during the pandemic. The magnitude of the mortality increases, both nationally and at the state level, were close to what would be expected if diabetes increased COVID-19 mortality by fivefold. Many studies have shown that diabetes increases the risk of COVID-19 mortality, with an effect size around 2 (*Williamson et al., 2020*; *Huang et al., 2020*; *Kastora et al., 2022*). Impaired immune function and chronic inflammation have been identified as mechanisms driving poor outcomes for diabetes patients (*Figueroa-Pizano et al., 2021*). The discrepancy between the observed excess and our expectations may come down to uncertainty in the SARS-CoV-2 infection rates assumed in our demographic model. The population living with diabetes is slightly younger than that of the other conditions (mean age, 58–60 years), while we used serologic infection rates reported for individuals over 65 years in our main analysis. The SARS-CoV-2 attack rate among those 50–64 years was 10.1% at the end of 2020, compared to 6.3% in individuals over 65 (*Centers for Disease Control and Prevention, 2023*). A sensitivity analysis using this higher attack rate in our demographic model

lends more support to the hypothesis that diabetes increases COVID-19 mortality by twofold, rather than fivefold as found in our main analysis.

Our second positive control, Alzheimer's, revealed surprising results. Although we observed significant excess mortality in MC Alzheimer's data, it was still less than expected under the null hypothesis that Alzheimer's was not a risk factor for COVID-19 mortality. This is unexpected in light of several observational studies that have shown Alzheimer's to be a risk factor (*Tahira et al., 2021*; *Wang et al., 2021*; *Zhang et al., 2021*). As with cancer and diabetes, there is uncertainty in the SARS-CoV-2 infection rates used in the demographic model, due to the potential effect of shielding and the age-specific SARS-CoV-2 infection risk of the Alzheimer's population. We estimated that the average age of the population living with Alzheimer's disease was 80–81 years, and the infection rates for the general population over 65 years may not accurately reflect exposure in this subpopulation. Decreasing the attack rates by 20–30% (down to 4.4–5.0%) puts the observed estimates in the range of the expectations under the null hypothesis. Overall, given uncertainty in SARS-CoV-2 attack rates and the age and size of the population-at-risk for all studied conditions, our demographic model projections are not an exact tool to titrate excess mortality nor the RR associated with each condition. Our model merely serves as an illustration of the role of demography and competing risks.

Most vital statistics studies of the COVID-19 pandemic have relied on UC-specific deaths, which are prone to changes in coding practices. Our initial hypothesis going into this work was that coding changes associated with a better recognition of the impact of SARS-CoV-2 led to an underestimation of excess mortality from cancer, affecting our perception of the relationship between cancer and COVID-19. We certainly found an effect of coding changes, where for instance a drop in excess mortality in underlying cancer deaths turned into an increase in MC (any-listed) cancer deaths, particularly in the first COVID-19 pandemic wave. A similar observation was made by *Fedeli et al., 2024*. The impact of coding changes was also seen in mortality from other chronic conditions but was particularly important for cancer. Yet both the absolute and relative excess mortality elevation remained modest for cancer, even after adjustment for coding changes, highlighting the importance of additional mechanisms such as competing mortality risks between COVID-19 and cancer.

An interesting hypothesis was put forward 20 years ago proposing that immunosuppression from cancer may explain the lack of excess cancer mortality in the 1968 influenza pandemic – the immune incompetence rescue hypothesis (*Reichert et al., 2004*). This hypothesis contends that it is a detrimental immune response that leads to influenza death. A similar hypothesis was put forward to explain the extreme mortality in young healthy adults in the 1918 pandemic (*Short et al., 2018*). However, observational studies have found that patients with hematological cancers have twice the risk of dying compared to patients without cancer, likely due to the immunosuppression associated with their malignancy and treatment (*Han et al., 2022a*; *Starkey et al., 2023*; *Williamson et al., 2020*). Under the immune incompetence rescue hypothesis, hematological cancers would be expected to have the lowest excess mortality of all types of cancers. Our excess mortality analysis reveals instead that hematological cancers were the most impacted by the pandemic, relative to other types of cancer, with observed mortality patterns consistent with a fivefold increase in risk of COVID-19 death in patients with hematological cancers. Overall, we do not find any support for the immune competence rescue hypothesis.

Our study is subject to limitations. First, we did not study the potential long-term consequences of the pandemic on cancer care, which may have resulted in avoidance of the healthcare system for diagnosis or treatment. We did not see any delayed pandemic effect on mortality from pancreatic cancer, which may have manifested in 2020 given the very low survival rate of this cancer (*Lemanska et al., 2023*), but we cannot rule out longer-term effects on breast or colorectal cancers that would not be seen until 2021 or later (*Doan et al., 2023*; *Han et al., 2023*; *Haribhai et al., 2023*; *Lee et al., 2023b*; *Nascimento de Lima et al., 2023*; *Nickson et al., 2023*; *Nonboe et al., 2023*; *Tope et al., 2023*). Interestingly, in the US, all-cause underlying cancer mortality rates do not appear to rise between 2020 and 2023 (*Appendix 1—figure 10*), but data prior to the pandemic show a rise in cancer incidence, largely driven by increasing cancer rates in younger adults (*Han et al., 2023*; *Siegel et al., 2024*). Additional years of data will be important to evaluate the long-term impacts of the COVID-19 pandemic and these changing demographics on cancer mortality rates. Additional years of data will also be important for assessing the impact of vaccination on the relationship between cancer and COVID-19; there is evidence that vaccines may be less immunogenic in patients with cancer compared

to those without (*Seneviratne et al., 2022*). Another limitation of our study is the reliance on mortality as an outcome, and not the risk of COVID-19-related hospitalization and morbidity, and Long COVID in cancer patients. A small US study reported that 60% of cancer patients suffered Long COVID symptoms (*Dagher et al., 2023*). Future analyses using hospitalization data and electronic medical records may provide additional insights on how different cancer stages or other comorbidities may contribute to increased risk of severe COVID-19 outcomes. Lastly, a few methodological limitations are worth raising. Though it was important to assess excess mortality in state-level data because of asynchrony in pandemic waves, confidence intervals in state-level estimates were large, particularly for specific types of cancers, affecting significance levels. Additional methodological limitations relate to our demographic model, especially as regards assumptions about SARS-CoV-2 infection rates in populations of different ages and with different chronic conditions. Importantly, our conclusions regarding the importance of competing risks are robust to these assumptions. Lastly, our study is a time-trend analysis and – like cohort and case-control studies – correlation does not necessarily imply causation. However, the intensity and brevity of COVID-19 pandemic waves in space and time lends support to our analyses.

## Conclusion

Our detailed excess mortality study considered six cancer types and found that there is at most a modest elevation in cancer mortality during the COVID-19 pandemic in the US. Our results demonstrate the importance of considering MC-of-death records to accurately reflect changes in coding practices associated with the emergence of a new pathogen. In contrast to earlier studies, we propose that lack of excess cancer mortality during the COVID-19 pandemic reflects the competing mortality risk from cancer (especially for deadly types like pancreatic and lung cancers) itself rather than protection conferred from immunosuppression. We note the more pronounced elevation in mortality from hematological cancers during the pandemic, compared to other cancers and to expectations from a demographic model, which aligns with a particular group of cancer patients singled out in several cohort studies. Future research on the relationship between COVID-19 and cancer should concentrate on additional outcomes, such as excess hospitalizations, Long COVID, changes in screening practices during COVID-19, and longer-term patterns in cancer mortality.

## Materials and methods

### Data sources

#### US National vital statistics

We obtained individual ICD-10-coded death certificate data with exact date of death from the US for the period January 1, 2014, to December 31, 2020. Each death certificate has one underlying cause (UC) of death, defined as the disease or injury that initiated the train of events leading directly to death, and up to 20 causes of death in total, referred to here as multiple-cause (MC) mortality. We considered 10 conditions, including diabetes, Alzheimer's disease, IHD, kidney disease, and six types of cancer (all-cause cancer, colorectal, breast, pancreatic, lung, and hematological; see *Table 1* and *Appendix 1—table 1* for a list of disease codes). We chose these types of cancer to illustrate conditions for which the 5-year survival rate is low (13% and 25%, respectively, for pancreatic and lung cancers) and high (65% and 91%, respectively, for colorectal and breast cancers) (*National Cancer Institute, 2024*). Hematological cancer (67% 5-year survival) was included because it was singled out as a risk factor by previous studies. We compiled time series by week, geography (aggregated National, NY, TX, and CA), and cause of death, separately for UC and MC mortality.

To observe longer-term trends in later years of the COVID-19 pandemic, we downloaded aggregated weekly level data from 2021 to 2023 for all-cause cancer, diabetes, and Alzheimer's disease from CDC Wonder.

#### Estimated populations living with each chronic condition

We estimated the size of the population-at-risk for all-cause and specific cancers using the 5-year limited duration prevalence estimates provided by the US Cancer Statistics webpage (*U.S. Cancer Statistics Working Group, 2023*). Estimates for diabetes were drawn from CDC's Behavioral Risk Factor Surveillance System Chronic Disease Indicators (CDC, Division of Population Health). Estimates

for Alzheimer's disease were taken from publications from the Alzheimer's Association (*Rajan et al., 2021*; *Dhana et al., 2023*).

For each condition, age-specific prevalence data were tabulated for the US and for each state separately. For cancer, age-level data were only available at the national level so these age-specific prevalence estimates were applied to the populations for each of the three states considered (NY, CA, TX). Age-level data were provided for all ages for cancer (<20 years, 20–80 years in 10-year groupings, ≥80 years), for adults ≥18 for diabetes (18–44 years, 45–64 years, ≥65 years), and for adults ≥65 for Alzheimer's disease (65–74 years, 75–84 years, ≥85 years). A weighted mean age for the population-at-risk for each condition was calculated using the mid-point for each age group.

## Other data sources

To compare vital statistics patterns with COVID-19 surveillance data, we accessed national and state counts of laboratory-confirmed COVID-19 cases in 2020, from the CDC (*Centers for Disease Control and Prevention, 2022*).

To clarify the expected role of COVID-19 on excess mortality, we compiled data on the proportion of the population with serologic evidence of SARS-CoV-2 infection from the CDC dashboard (*Centers for Disease Control and Prevention, 2023*). We further compiled data on estimated age-specific IFRs from COVID-19, provided by single year of age (*COVID-19 Forecasting Team, 2022*).

## Statistical approach

### Weekly excess mortality models

Similar to other influenza and COVID-19 excess mortality studies (*Islam et al., 2021*; *Karlinsky and Kobak, 2021*; *Lee et al., 2023a*; *Msemburi et al., 2023*), we established a predicted baseline of expected mortality for each time series, and computed the excess mortality as the excess in observed deaths over this baseline. To establish baselines for each disease nationally and in each state, we applied negative binomial regression models to weekly mortality counts for each cause of death, smoothed with a 5-week moving average and rounded to the nearest integer. Models included harmonic terms for seasonality, time trends, and an offset for population size. For each condition and location, we used Akaike information criterion (AIC) to select between three models with different time trends (see Appendix 1 - Supplemental Methods, *Appendix 1—figure 11*, for the final model selection for each location and condition), following:

Model 1:

Weekly_mortality = t + cos(2πt/52.17)+sin(2πt/52.17)+offset(log(population)), where t represents week.

Model 2:

Weekly_mortality = t + t$^2$+cos(2πt/52.17)+sin(2πt/52.17)+offset(log(population)), where t represents week.

Model 3:

Weekly_mortality = t + t2+t3+cos(2πt/52.17)+sin(2πt/52.17)+offset(log(population)), where t represents week.

We fitted national- and state-level models for each mortality outcome from January 19, 2014, to March 1, 2020, and projected the baseline forward until December 6, 2020, the last complete week of smoothed mortality data. Models were fitted using the MASS package in R version 4.3.

Using COVID-19-coded death certificates from March 1, 2020, to December 6, 2020, we established the timing of each pandemic wave from trough to trough. We found that nationally, the first wave occurred from March 1, 2020, to June 27, 2020; the second wave from June 28, 2020, to October 3, 2020, and the third from October 4, 2020, to December 6, 2020 (the third wave was not completed by the last week of available smoothed data on December 6, 2020). For NY, the pandemic pattern was characterized by an intense first wave in spring 2020, while TX had its major wave in summer 2020 and CA in late 2020. Comparison of mortality patterns from these three states provides an opportunity to separate the effect of SARS-CoV-2 infection from that of behavioral changes later in the pandemic. For instance, the effects of healthcare avoidance would predominate in CA or TX in spring 2020, as there was little SARS-CoV-2 activity but much media attention on COVID-19, with cancer patients potentially avoiding medical care out of fear of getting infected. In contrast, risk of infection would

dominate in NY in spring 2020, and behavioral factors may only play a role as SARS-CoV-2 awareness increased and the wave was brought under control by social distancing.

We estimated weekly excess mortality by subtracting the predicted baseline from the observed mortality. We summed weekly estimates to calculate excess mortality for the full pandemic period and for each of the three waves within the first year of the pandemic. In addition to estimating the absolute effects of the pandemic on mortality, we also calculated relative effects by dividing excess deaths in each diagnosis group by the model baseline. Confidence intervals on excess mortality estimates were calculated by resampling the estimated model coefficients 10,000 times using a multivariate normal distribution and accounting for negative binomial errors in weekly mortality counts.

We used Pearson correlation to test synchronicity patterns in weekly excess mortality from different cancers and chronic conditions to underlying COVID-19 deaths. Correlation analysis assumes a direct and immediate effect of COVID-19 on cancer mortality. We also investigated the possibility of delayed effects or harvesting by inspecting the time series for evidence of such effects and by comparing total excess deaths for distinct pandemic waves and the whole of 2020.

## Projections of excess mortality under the null hypothesis of no specific COVID-19 mortality risk of each condition

To further test the impact of age on the association between chronic conditions and COVID-19 and clarify the additional risk due to each chronic condition, we projected the number of COVID-19 deaths under the null hypothesis that demographic characteristics alone (size, age, and baseline mortality risk for each condition) are driving excess mortality, and that there is no interaction between the condition and SARS-CoV-2 infection. Excess mortality projections were then compared with observed excess mortality. We only used MC deaths for this approach to account for the possibility that some individuals may suffer from multiple conditions. For example, an estimated 11.5% of US adults with type 2 diabetes also have a history of cancer (*Yeh et al., 2018*).

We first calculated the number of expected COVID-19 infections among persons living with a certain chronic condition, by multiplying the estimated number of individuals living with the condition by the reported SARS-CoV-2 seroprevalence among individuals ≥65 years at specific time points during 2020. For the national data and CA, we used results from the survey conducted from November 23 to December 12, 2020. For NY we used estimates from the survey conducted from July 27 to August 13, 2020 (the earliest data available). And for TX we used the survey conducted from October 5–19, 2020 (following the large summer wave) (*Centers for Disease Control and Prevention, 2023*). We then multiplied this by the COVID-19 IFR based on the estimated mean age of individuals living with the condition (*COVID-19 Forecasting Team, 2022*) to arrive at the projected number of COVID-19-related excess deaths for a particular condition during 2020. We put uncertainty intervals around these estimates using the lower and upper bounds from the estimated attack rates and COVID-19 IFRs.

To obtain a relative metric of expected COVID-19 burden, we divided projected COVID-19 excess deaths by total deaths in each diagnosis group in the 2019 baseline period (March to December 2019, for the national data. For the states we used the months in 2019 corresponding to their large waves in 2020), resulting in an expected percentage elevation over baseline in 2020. We compared this null expectation to the observed percentage elevation over baseline from our excess mortality models. We also generated the expected number of excess deaths under alternative hypotheses where each condition is associated with a two- or fivefold increased risk of COVID-19-related death given infection (i.e. the baseline age-adjusted IFR used in the null hypothesis was increased two- or fivefold).

The equation for the expected percent increase in excess mortality over baseline deaths under the null hypothesis, for a specific risk condition (cancer, diabetes, Alzheimer) and time period, can be written as:

Expected percent increase in excess mortality for a chronic condition and time period = (size of population-at-risk for the condition * SARS-CoV-2 infection rate for the period * age-specific IFR)/ baseline mortality for the condition in comparable period in 2019.

The expected mortality increases under the alternative hypothesis of a two- or fivefold increased risk of COVID-19 death from the condition under study is modeled by multiplying the right-hand side of the above equation by the increased risk (i.e. we assume that presence of the underlying condition will increase the IFR by two- or fivefold compared to the IFR for the general population).

## Acknowledgements

This paper is dedicated to our colleague Robert J Taylor who succumbed to cancer in 2022 and who wanted to know if a cancer diagnosis was a COVID-19 mortality risk factor. LS acknowledges funding from the Carlsberg Foundation, grant number CF20-0046. LS and CLH acknowledge funding from Danish National Research Foundation (grant number DNRF170) for the PandemiX Center of Excellence. CLH has received contract-based hourly consulting fees from Sanofi outside of the submitted work.

## Additional information

### Funding

| Funder | Grant reference number | Author |
| --- | --- | --- |
| Carlsberg Foundation | CF20-0046 | Lone Simonsen |
| Danish National Research Foundation | DNRF170 | Chelsea L Hansen Lone Simonsen |

The funders had no role in study design, data collection and interpretation, or the decision to submit the work for publication.

### Author contributions

Chelsea L Hansen, Data curation, Formal analysis, Visualization, Methodology, Writing – original draft, Writing – review and editing; Cécile Viboud, Lone Simonsen, Conceptualization, Data curation, Formal analysis, Supervision, Visualization, Methodology, Writing – original draft, Writing – review and editing

### Author ORCIDs

Chelsea L Hansen ⓘ https://orcid.org/0000-0002-4526-6772
Cécile Viboud ⓘ https://orcid.org/0000-0003-3243-4711
Lone Simonsen ⓘ https://orcid.org/0000-0003-1535-8526

Reviewer #1 (Public Review): https://doi.org/10.7554/eLife.93758.3.sa1
Author response https://doi.org/10.7554/eLife.93758.3.sa2

## Additional files

### Supplementary files
• MDAR checklist

### Data availability

Individual-level mortality data with exact date of death were obtained from the National Center for Healthcare Statistics (NCHS). Individual-level data with exact date of death are not publicly available due to privacy concerns, but descriptive characteristics have been summarized in *Table 1* and *Appendix 1—table 1*. Researchers wishing to access these data must submit an application to NCHS following the instructions provided here: https://www.cdc.gov/nchs/nvss/nvss-restricted-data. htm. Individual-level data without the exact data of death are publicly available and can be downloaded from here: https://www.cdc.gov/nchs/data_access/vitalstatsonline.htm. All analyses shown in this paper can be replicated based on the aggregated data and code posted at the following public GitHub repository: https://github.com/chelsea-hansen/Disentangling-the-relationship-between-cancer-mortality-and-COVID-19 (copy archived at *Hansen, 2024*). Additional weekly, aggregated mortality data for trends in cancer, diabetes, and Alzheimer's mortality post-2020 are publicly available through CDC Wonder and have been included in the GitHub repository. Data used for the demographic model were gathered from publicly available sources. These data, along with the code for the model, have also been posted to the GitHub repository. Weekly, state-level data on recorded COVID-19 cases and deaths are publicly available. Data were downloaded from here: https://data.

cdc.gov/Case-Surveillance/Weekly-United-States-COVID-19-Cases-and-Deaths-by-/pwn4-m3yp and have also been posted as a .csv file to the GitHub repository.

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

## Appendix 1

### Supplemental methods

#### Model selection and cross-validation

Time series models included harmonic terms for seasonality, time trends, and an offset for population size. For each condition and location, we used AIC to select between three models with different time trends. The starting model (Model 1) included only a linear time trend. We then tested this against a model with linear and quadratic time trends (Model 2). If the AIC of Model 2 was not 2 less than Model 1, Model 1 was used as the final model. If the AIC of Model 2 was 2 less than Model 1, then Model 2 was tested against a model with linear, quadratic, and cubic time trends (Model 3). If the AIC of Model 3 was not 2 less than Model 2, then Model 2 was taken as the final model. If the AIC of Model 3 was 2 less than Model 2, Model 3 was taken as the final model. The final model for each condition and location was then applied to the data from 2014 to 2018 only and used to predict the 2019 data. The coverage probability was calculated as the proportion of weeks of observed data in 2019 that fell within the 95% prediction interval of the time series model. The final model selected for each condition and location is provided in *Appendix 1—figure 11*.

#### Characteristics of cancer, diabetes, and Alzheimer's deaths in the pre-pandemic period

For each chronic condition studied (cancer, diabetes, Alzheimer's), we assessed potential changes in the characteristics of deaths during the pandemic period that are unrelated to timing but may signal an association with COVID-19. For instance, age is known to be a major risk factor for COVID-19 mortality. For each chronic condition, we computed the average age-at-death in the pre-pandemic year 2019, and compared this to the average age-at-death in 2020. The second potential confounder is living arrangement, as individuals living in nursing homes may be at increased risk of exposure (and death) to COVID-19 due to mixing, even if their underlying condition is not per se a risk factor. To test this hypothesis, we also compared the proportion of individuals in each disease group who died in nursing homes in 2019 and 2020. And finally, to illustrate the impact of coding practices we compared ICD-10 letter categories between 2020 and 2019 for the UC of death when cancer or diabetes are included on the death certificate but are not listed as the UC of death (*Appendix 1—figure 9*). For 2020, we further compared death certificates listing both COVID-19 and cancer to those listing both COVID-19 and diabetes. For all comparisons between 2019 and 2020 data are limited to March to December to isolate the pandemic period.

**Appendix 1—table 1.** Diagnosis groups and corresponding ICD-10 codes, number of underlying and multiple cause deaths, mean age in years at time of death, the percentage of deaths occurring at home, and the percentage of deaths occurring in nursing homes for 2019 and 2020.

| Year | Diagnosis group | ICD-10 codes | Underlying cause | | | | Multiple cause | | | |
| | | | No. deaths | Mean age, years (IQR) | %Home/ ER | %Nursing home | No. deaths | Mean age, years (IQR) | %Home/ ER | %Nursing home |
|---|---|---|---|---|---|---|---|---|---|---|
| 2019 | Cancer | C00-C99 | 493,397 | 72 (64–81) | 45 | 12 | 546,453 | 72 (64–82) | 44 | 13 |
| | Pancreatic cancer | C25 | 37,864 | 72 (64–80) | 51 | 9 | 39,798 | 72 (64–80) | 50 | 9 |
| | Lung cancer | C34 | 114,552 | 72 (65–80) | 45 | 12 | 123,622 | 72 (65–80) | 44 | 12 |
| | Colorectal cancer | C18-C20 | 42,484 | 71 (61–82) | 46 | 13 | 49,053 | 72 (62–83) | 45 | 14 |
| | Breast cancer | C50 | 35,115 | 69 (59–81) | 44 | 13 | 43,519 | 71 (61–83) | 43 | 15 |
| | Hematological cancer | C81-C96 | 47,174 | 74 (67–84) | 35 | 11 | 57,892 | 74 (67–84) | 35 | 12 |
| | Diabetes | E10-E14 | 70,763 | 72 (63–82) | 53 | 17 | 229,326 | 74 (65–84) | 46 | 19 |
| | Alzheimer's | G30 | 98,675 | 87 (82–92) | 29 | 50 | 118,993 | 87 (82–92) | 29 | 48 |
| | Ischemic heart disease | I20-I25 | 292,659 | 77 (67–88) | 50 | 18 | 440,225 | 77 (68–87) | 47 | 18 |

*Appendix 1—table 1 Continued on next page*

*Appendix 1—table 1 Continued*

| | | | Underlying cause | | | | Multiple cause | | | |
|---|---|---|---|---|---|---|---|---|---|---|
| | Kidney disease | N00-07, 17–19,25-28 | 46,120 | 76 (68–87) | 25 | 18 | 189,938 | 76 (67–87) | 20 | 15 |
| 2020 | Cancer | C00-C99 | 513,275 | 72 (64–81) | 55 | 8 | 586,503 | 72 (64–82) | 52 | 9 |
| | Pancreatic cancer | C25 | 39,893 | 72 (65–80) | 61 | 6 | 42,383 | 72 (65–80) | 60 | 6 |
| | Lung cancer | C34 | 115,554 | 72 (65–80) | 54 | 8 | 127,671 | 72 (65–80) | 53 | 8 |
| | Colorectal cancer | C18-C20 | 43,990 | 71 (61–82) | 56 | 9 | 52,319 | 72 (62–83) | 53 | 10 |
| | Breast cancer | C50 | 36,296 | 70 (60–81) | 54 | 10 | 47,094 | 72 (62–83) | 51 | 12 |
| | Hematological cancer | C81-C96 | 49,161 | 74 (67–84) | 46 | 8 | 64,840 | 74 (68–84) | 43 | 9 |
| | Diabetes | E10-E14 | 88,124 | 71 (62–82) | 58 | 15 | 343,061 | 73 (65–83) | 45 | 16 |
| | Alzheimer's | G30 | 115,256 | 86 (82–92) | 33 | 46 | 151,206 | 86 (82–92) | 31 | 47 |
| | Ischemic heart disease | I20-I25 | 327,854 | 76 (67–88) | 54 | 16 | 533,204 | 77 (68–87) | 49 | 16 |
| | Kidney disease | N00-07, 17–19,25-28 | 49,796 | 76 (68–87) | 30 | 15 | 255,708 | 75 (67–86) | 21 | 12 |

**Appendix 1—table 2.** Estimated Excess Deaths by Cause and Wave (National).
Estimated number of excess deaths and the percentage over baseline for each diagnosis group (National). Estimates are aggregated over all of 2020 and for each COVID-19 wave during 2020.

| Cause of death | Wave | Multiple cause | | Underlying cause | |
|---|---|---|---|---|---|
| | | Excess deaths | % Over baseline | Excess deaths | % Over baseline |
| Cancer | Overall | 13,601* | 3.0 | 11 | 0.0 |
| | 1 | 79 | 0.0 | –3917* | –2.0 |
| | 2 | 6519* | 4.0 | 2662 | 2.0 |
| | 3 | 7003* | 6.0 | 1266 | 1.0 |
| Pancreatic cancer | Overall | –25 | –0.0 | –282 | –1.0 |
| | 1 | –213 | –1.0 | –281 | –2.0 |
| | 2 | 44 | 0.0 | –30 | –0.0 |
| | 3 | 144 | 1.0 | 29 | 0.0 |
| Lung cancer | Overall | 1102 | 1.0 | –814 | –1.0 |
| | 1 | –729 | –1.0 | –1221 | –3.0 |
| | 2 | 784 | 2.0 | 249 | 1.0 |
| | 3 | 1047 | 4.0 | 158 | 1.0 |
| Breast cancer | Overall | 838 | 2.0 | –438 | –1.0 |
| | 1 | –66 | –0.0 | –415 | –3.0 |
| | 2 | 437 | 3.0 | 81 | 1.0 |
| | 3 | 467 | 5.0 | –105 | –1.0 |
| Colorectal cancer | Overall | 988 | 2.0 | –168 | –0.0 |
| | 1 | –169 | –1.0 | –463 | –3.0 |
| | 2 | 454 | 3.0 | 112 | 1.0 |
| | 3 | 703* | 6.0 | 183 | 2.0 |

*Appendix 1—table 2 Continued on next page*

*Appendix 1—table 2 Continued*

| Cause of death | Wave | Multiple cause | | Underlying cause | |
|---|---|---|---|---|---|
| | | Excess deaths | % Over baseline | Excess deaths | % Over baseline |
| Hematological cancers | Overall | 3615* | 7.0 | 111 | 0.0 |
| | 1 | 546 | 2.0 | –447 | –2.0 |
| | 2 | 1412* | 8.0 | 412 | 3.0 |
| | 3 | 1657* | 12.0 | 146 | 1.0 |
| Diabetes | Overall | 82,318* | 37.0 | 10,784* | 16.0 |
| | 1 | 25,306* | 25.0 | 2305* | 7.0 |
| | 2 | 27,534* | 38.0 | 4330* | 20.0 |
| | 3 | 29,477* | 56.0 | 4148* | 26.0 |
| Alzheimer's | Overall | 21,712* | 19.0 | 8528* | 9.0 |
| | 1 | 4763* | 9.0 | 547 | 1.0 |
| | 2 | 8054* | 22.0 | 4257* | 14.0 |
| | 3 | 8894* | 33.0 | 3724* | 16.0 |
| Ischemic heart disease | Overall | 58,793* | 14.0 | 17,194* | 6.0 |
| | 1 | 12,042* | 6.0 | 862 | 1.0 |
| | 2 | 21,783* | 16.0 | 7912* | 9.0 |
| | 3 | 24,967* | 25.0 | 8419* | 13.0 |
| Kidney disease | Overall | 41,907* | 22.0 | 785 | 2.0 |
| | 1 | 8182* | 10.0 | –1048 | –5.0 |
| | 2 | 14,767* | 25.0 | 777 | 5.0 |
| | 3 | 18,958* | 44.0 | 1056* | 10.0 |

*Confidence interval does not include zero.

**Appendix 1—table 3.** Estimated Excess Deaths by Cause and Wave (New York).
Estimated number of excess deaths and the percentage over baseline for each diagnosis group (New York). Estimates are aggregated over all of 2020 and for each COVID-19 wave during 2020.

| Cause of death | Wave | Multiple cause | | Underlying cause | |
|---|---|---|---|---|---|
| | | Excess deaths | % Over baseline | Excess deaths | % Over baseline |
| Cancer | Overall | 1012 | 4.0 | –557 | –2.0 |
| | 1 | 747 | 6.0 | –474 | –5.0 |
| | 2 | 120 | 1.0 | -6 | –0.0 |
| | 3 | 144 | 2.0 | –77 | –1.0 |
| Pancreatic cancer | Overall | –29 | –1.0 | –58 | –3.0 |
| | 1 | 8 | 1.0 | –16 | –2.0 |
| | 2 | -1 | –0.0 | -9 | –1.0 |
| | 3 | –37 | –6.0 | –33 | –6.0 |
| Lung cancer | Overall | 47 | 1.0 | –163 | –3.0 |
| | 1 | 27 | 1.0 | –143 | –7.0 |
| | 2 | 23 | 1.0 | 16 | 1.0 |
| | 3 | -3 | –0.0 | –36 | –3.0 |

*Appendix 1—table 3 Continued on next page*

*Appendix 1—table 3 Continued*

| Cause of death | Wave | Multiple cause | | Underlying cause | |
|---|---|---|---|---|---|
| | | Excess deaths | % Over baseline | Excess deaths | % Over baseline |
| Breast cancer | Overall | 205 | 9.0 | –46 | –2.0 |
| | 1 | 151 | 16.0 | –34 | –4.0 |
| | 2 | 31 | 4.0 | 3 | 0.0 |
| | 3 | 23 | 4.0 | –15 | –3.0 |
| Colorectal cancer | Overall | 189 | 8.0 | 42 | 2.0 |
| | 1 | 91 | 9.0 | –16 | –2.0 |
| | 2 | 40 | 5.0 | 26 | 4.0 |
| | 3 | 58 | 9.0 | 33 | 6.0 |
| Hematological cancers | Overall | 156 | 5.0 | –149 | –6.0 |
| | 1 | 121 | 10.0 | –107 | –11.0 |
| | 2 | 1 | 0.0 | –25 | –3.0 |
| | 3 | 35 | 5.0 | –18 | –3.0 |
| Diabetes | Overall | 7240* | 66.0 | 866* | 26.0 |
| | 1 | 5945* | 128.0 | 568* | 40.0 |
| | 2 | 631* | 18.0 | 121 | 11.0 |
| | 3 | 664* | 24.0 | 177 | 21.0 |
| Alzheimer's | Overall | 884* | 26.0 | 233 | 9.0 |
| | 1 | 734* | 49.0 | 188 | 16.0 |
| | 2 | 1 | 0.0 | 1 | 0.0 |
| | 3 | 150 | 17.0 | 44 | 6.0 |
| Ischemic heart disease | Overall | 7118* | 25.0 | 3756* | 17.0 |
| | 1 | 6607* | 54.0 | 4092* | 44.0 |
| | 2 | 179 | 2.0 | –184 | –3.0 |
| | 3 | 331 | 5.0 | –152 | –3.0 |
| Kidney disease | Overall | 2438* | 34.0 | 51 | 3.0 |
| | 1 | 1946* | 63.0 | 22 | 3.0 |
| | 2 | 144 | 6.0 | –13 | –2.0 |
| | 3 | 349* | 19.0 | 42 | 8.0 |

*Confidence interval does not include zero.

**Appendix 1—table 4.** Estimated Excess Deaths by Cause and Wave (Texas).
Estimated number of excess deaths and the percentage over baseline for each diagnosis group (Texas). Estimates are aggregated over all of 2020 and for each COVID-19 wave during 2020.

| Cause of death | Wave | Multiple cause | | Underlying cause | |
|---|---|---|---|---|---|
| | | Excess deaths | % Over baseline | Excess deaths | % Over baseline |
| Cancer | Overall | 602 | 2.0 | –130 | –0.0 |
| | 1 | –48 | –0.0 | –62 | –0.0 |
| | 2 | 467 | 4.0 | 39 | 0.0 |
| | 3 | 183 | 2.0 | –107 | –1.0 |

*Appendix 1—table 4 Continued on next page*

*Appendix 1—table 4 Continued*

| Cause of death | Wave | Multiple cause | | Underlying cause | |
|---|---|---|---|---|---|
| | | Excess deaths | % Over baseline | Excess deaths | % Over baseline |
| Pancreatic cancer | Overall | 1 | 0.0 | 5 | 0.0 |
| | 1 | −36 | −3.0 | −36 | −4.0 |
| | 2 | 17 | 2.0 | 24 | 3.0 |
| | 3 | 20 | 3.0 | 17 | 3.0 |
| Lung cancer | Overall | 176 | 2.0 | 108 | 2.0 |
| | 1 | 33 | 1.0 | 31 | 1.0 |
| | 2 | 60 | 2.0 | 27 | 1.0 |
| | 3 | 84 | 5.0 | 49 | 3.0 |
| Breast cancer | Overall | −19 | −1.0 | −131 | −5.0 |
| | 1 | −54 | −4.0 | −54 | −6.0 |
| | 2 | 29 | 3.0 | −25 | −3.0 |
| | 3 | 6 | 1.0 | −51 | −8.0 |
| Colorectal cancer | Overall | −12 | −0.0 | −92 | −3.0 |
| | 1 | −33 | −2.0 | −49 | −4.0 |
| | 2 | 4 | 0.0 | −34 | −3.0 |
| | 3 | 17 | 2.0 | −10 | −1.0 |
| Hematological cancers | Overall | 194 | 5.0 | −12 | −0.0 |
| | 1 | 24 | 2.0 | 1 | 0.0 |
| | 2 | 136 | 11.0 | 21 | 2.0 |
| | 3 | 33 | 3.0 | −34 | −4.0 |
| Diabetes | Overall | 8902* | 49.0 | 618 | 11.0 |
| | 1 | 1411* | 19.0 | 61 | 3.0 |
| | 2 | 4612* | 77.0 | 420* | 23.0 |
| | 3 | 2879* | 62.0 | 138 | 9.0 |
| Alzheimer's | Overall | 2242* | 24.0 | 1184 | 15.0 |
| | 1 | 309 | 8.0 | 197 | 6.0 |
| | 2 | 1398* | 45.0 | 805* | 31.0 |
| | 3 | 536* | 21.0 | 181 | 8.0 |
| Ischemic heart disease | Overall | 6018* | 20.0 | 1700 | 9.0 |
| | 1 | 736 | 6.0 | 99 | 1.0 |
| | 2 | 3376* | 34.0 | 1228* | 19.0 |
| | 3 | 1905* | 24.0 | 374 | 7.0 |
| Kidney disease | Overall | 6724* | 47.0 | 579 | 19.0 |
| | 1 | 886* | 15.0 | 115 | 9.0 |
| | 2 | 3535* | 76.0 | 285* | 28.0 |
| | 3 | 2303* | 66.0 | 179 | 23.0 |

*Confidence interval does not include zero.

**Appendix 1—table 5.** Estimated Excess Deaths by Cause and Wave (California).
Estimated number of excess deaths and the percentage over baseline for each diagnosis group (California). Estimates are aggregated over all of 2020 and for each COVID-19 wave during 2020.

| Cause of death | Wave | Multiple cause | | Underlying cause | |
|---|---|---|---|---|---|
| | | Excess deaths | % Over baseline | Excess deaths | % Over baseline |
| Cancer | Overall | 991 | 2.0 | –29 | –0.0 |
| | 1 | –102 | –1.0 | –236 | –1.0 |
| | 2 | 564 | 3.0 | 125 | 1.0 |
| | 3 | 529 | 4.0 | 82 | 1.0 |
| Pancreatic cancer | Overall | –97 | –3.0 | –126 | –4.0 |
| | 1 | –28 | –2.0 | –39 | –3.0 |
| | 2 | –69 | –5.0 | –70 | –6.0 |
| | 3 | 0 | 0.0 | –18 | –2.0 |
| Lung cancer | Overall | –10 | –0.0 | –132 | –2.0 |
| | 1 | –82 | –3.0 | –96 | –3.0 |
| | 2 | 18 | 1.0 | –48 | –2.0 |
| | 3 | 54 | 3.0 | 13 | 1.0 |
| Breast cancer | Overall | 67 | 2.0 | –22 | –1.0 |
| | 1 | –44 | –3.0 | –34 | –3.0 |
| | 2 | 92 | 6.0 | 44 | 4.0 |
| | 3 | 20 | 2.0 | –33 | –4.0 |
| Colorectal cancer | Overall | 100 | 2.0 | 20 | 1.0 |
| | 1 | 7 | 0.0 | –4 | –0.0 |
| | 2 | 66 | 4.0 | 25 | 2.0 |
| | 3 | 27 | 2.0 | –1 | –0.0 |
| Hematological cancers | Overall | 279 | 5.0 | 52 | 1.0 |
| | 1 | 0 | 0.0 | –33 | –2.0 |
| | 2 | 164 | 9.0 | 64 | 4.0 |
| | 3 | 114 | 8.0 | 20 | 2.0 |
| Diabetes | Overall | 9163* | 39.0 | 1408* | 20.0 |
| | 1 | 1843* | 18.0 | 213 | 7.0 |
| | 2 | 3846* | 49.0 | 620* | 27.0 |
| | 3 | 3474* | 59.0 | 575* | 33.0 |
| Alzheimer's | Overall | 2143* | 14.0 | 594 | 5.0 |
| | 1 | 375 | 6.0 | –76 | –1.0 |
| | 2 | 1041* | 20.0 | 410 | 9.0 |
| | 3 | 726* | 18.0 | 259 | 8.0 |
| Ischemic heart disease | Overall | 5905* | 16.0 | 2888* | 11.0 |
| | 1 | 650 | 4.0 | 104 | 1.0 |
| | 2 | 2966* | 24.0 | 1581* | 19.0 |
| | 3 | 2289* | 25.0 | 1204* | 19.0 |

*Appendix 1—table 5 Continued on next page*

*Appendix 1—table 5 Continued*

| Cause of death | Wave | Multiple cause | | Underlying cause | |
|---|---|---|---|---|---|
| | | Excess deaths | % Over baseline | Excess deaths | % Over baseline |
| Kidney disease | Overall | 3858* | 21.0 | 8 | 0.0 |
| | 1 | 301 | 4.0 | −114 | −8.0 |
| | 2 | 1967* | 33.0 | 63 | 6.0 |
| | 3 | 1590* | 36.0 | 59 | 7.0 |

*Confidence interval does not include zero.

**National**

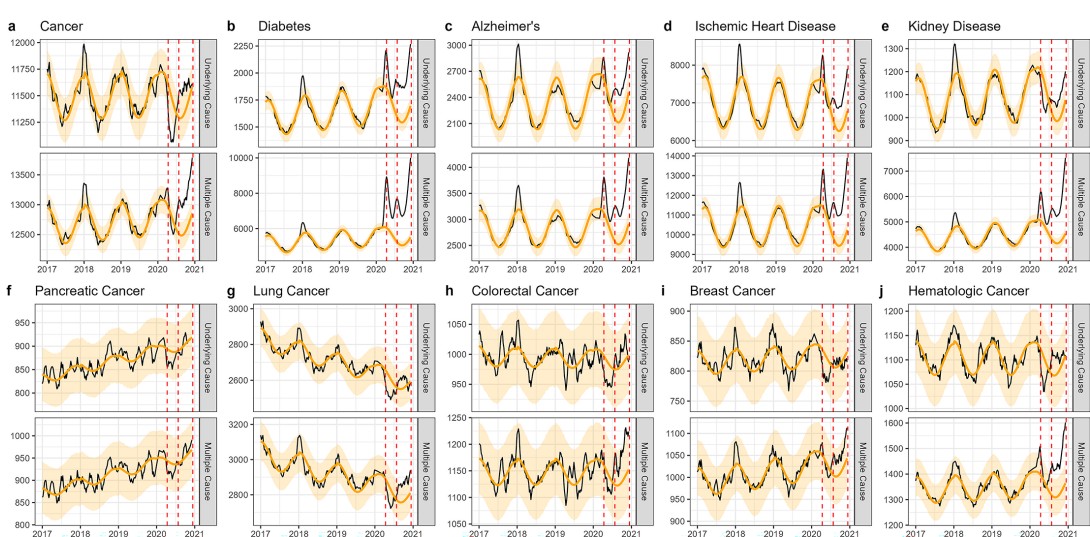

**Appendix 1—figure 1.** National-level weekly observed and estimated baseline mortality for each diagnosis group (Cancer (**a**), Diabetes (**b**), Alzheimer's (**c**), Ischemic Heart Disease (**d**), Kidney Disease (**e**), Pancreatic Cancer (**f**), Lung Cancer (**g**), Colorectal Cancer (**h**), Breast Cancer (**i**), Hematologica Cancer (**j**)) as both the underlying cause or anywhere on the death certificate (multiple cause) from 2017 to 2020. Red dashed lines represent the timing of the peaks for the three COVID-19 waves in 2020. Baselines during the pandemic are projected based on the previous years of data.

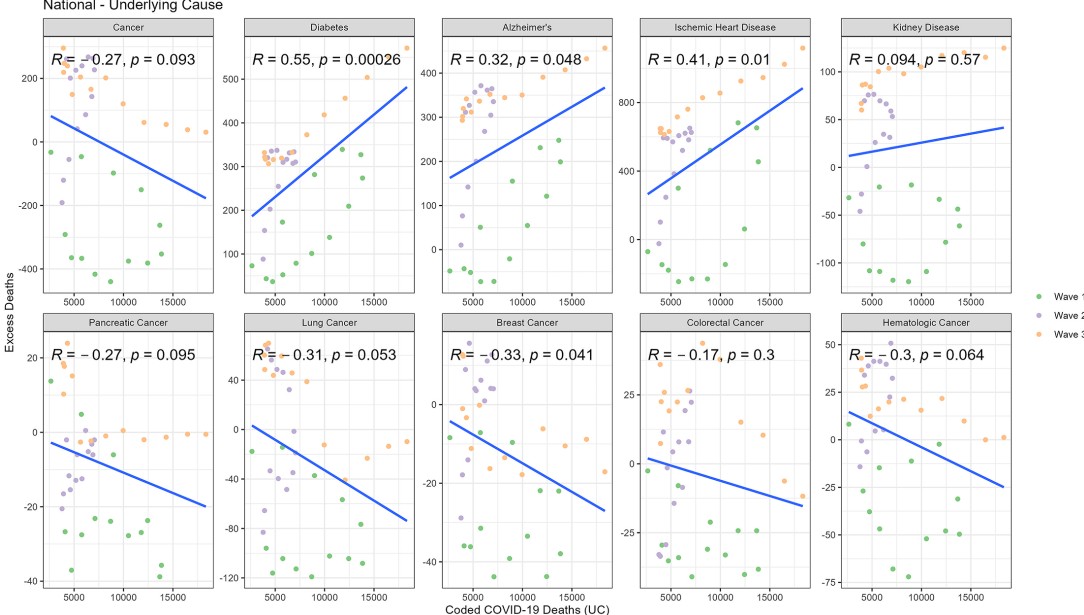

**Appendix 1—figure 2.** Correlation between weekly number of COVID-19-coded deaths and excess underlying deaths for each diagnosis group (National).

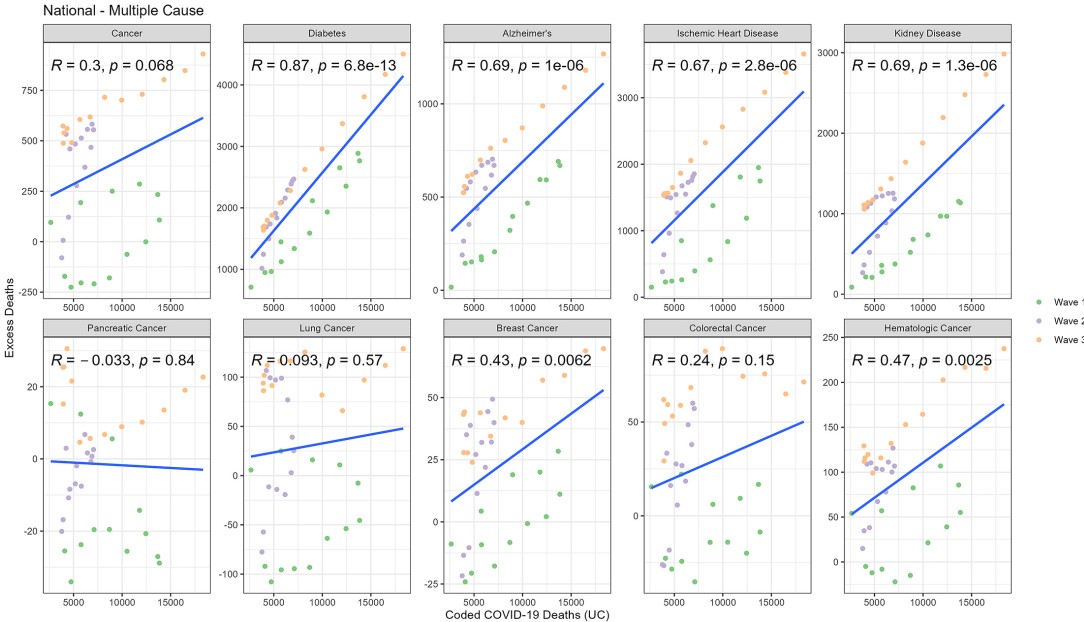

**Appendix 1—figure 3.** Correlation between weekly number of COVID-19-coded deaths and excess multiple cause deaths for each diagnosis group (National).

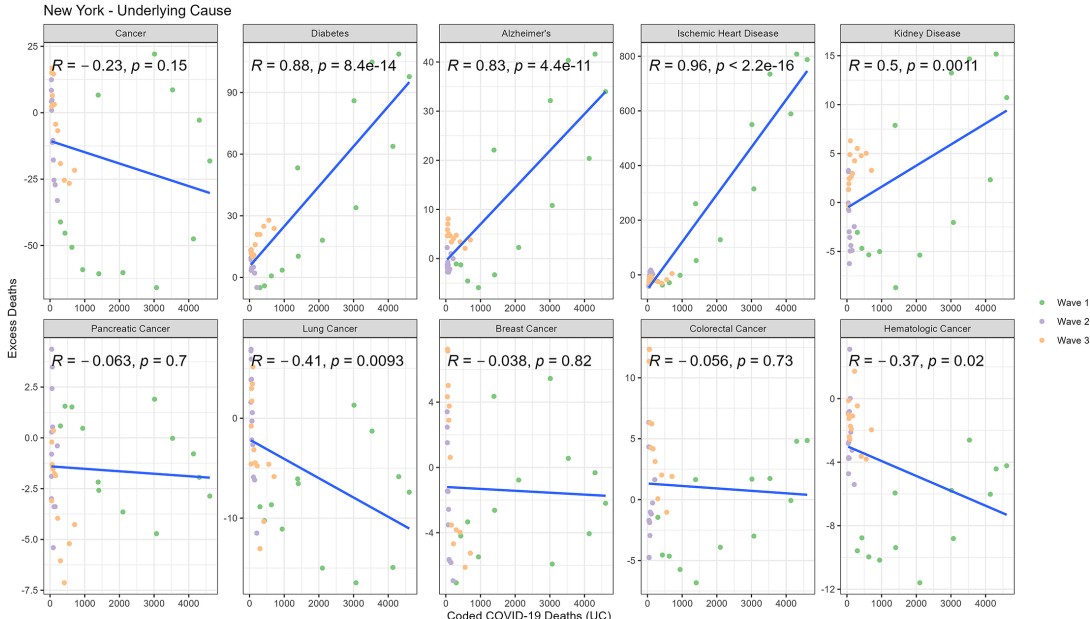

**Appendix 1—figure 4.** Correlation between weekly number of COVID-19-coded deaths and excess underlying deaths for each diagnosis group (New York).

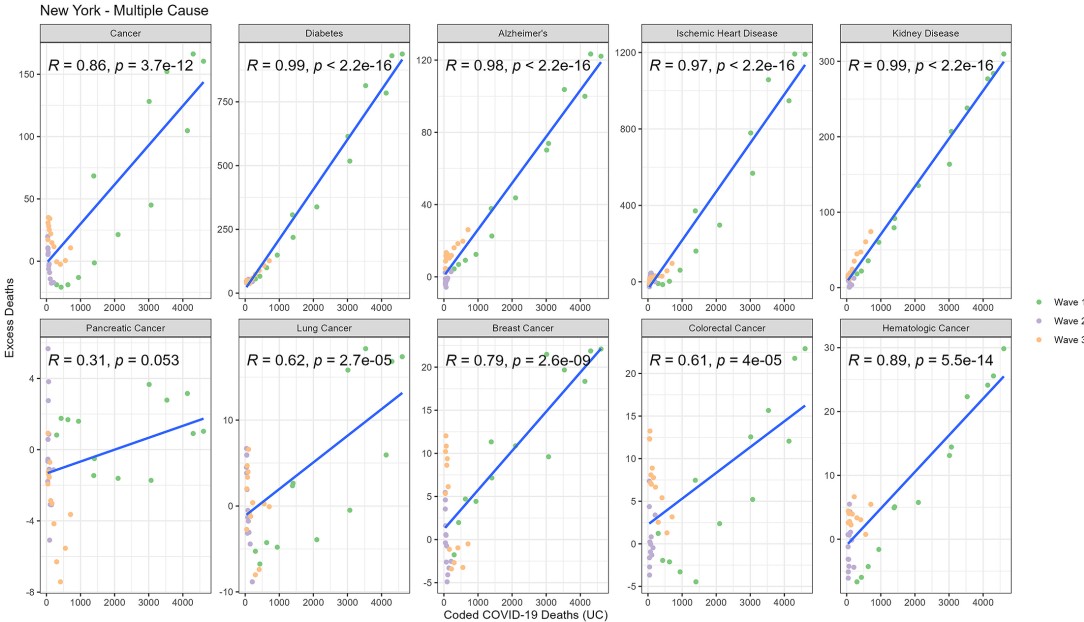

**Appendix 1—figure 5.** Correlation between weekly number of COVID-19-coded deaths and excess underlying deaths for each diagnosis group (New York).

**New York**

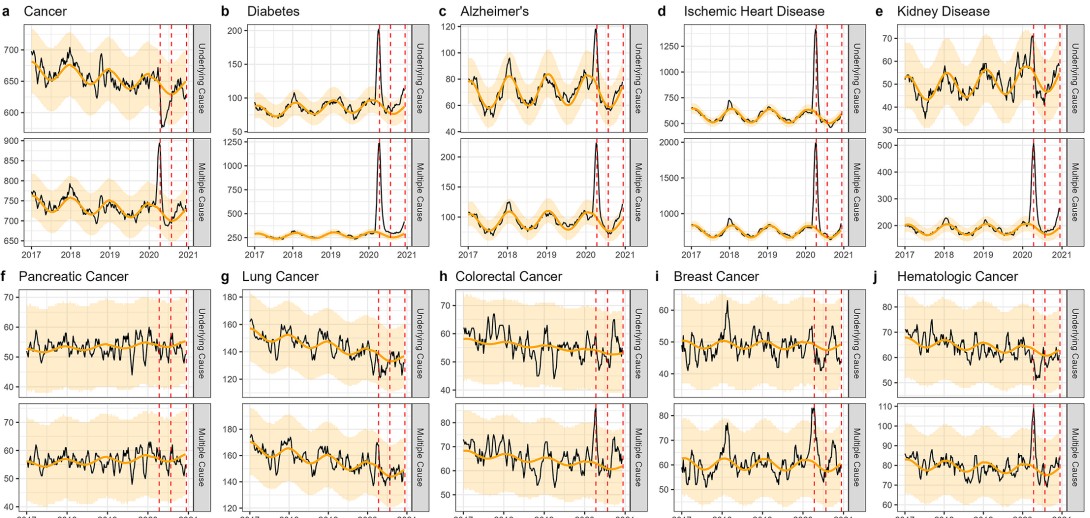

**Appendix 1—figure 6.** Weekly observed and estimated baseline mortality for each diagnosis group (Cancer (**a**), Diabetes (**b**), Alzheimer's (**c**), Ischemic Heart Disease (**d**), Kidney Disease (**e**), Pancreatic Cancer (**f**), Lung Cancer (**g**), Colorectal Cancer (**h**), Breast Cancer (**i**), Hematologica Cancer (**j**)) as both the underlying cause or anywhere on the death certificate (multiple cause) from 2017 to 2020 in New York. Red dashed lines represent the timing of the peaks for the three COVID-19 waves in 2020. Baselines during the pandemic are projected based on the previous years of data.

**Texas**

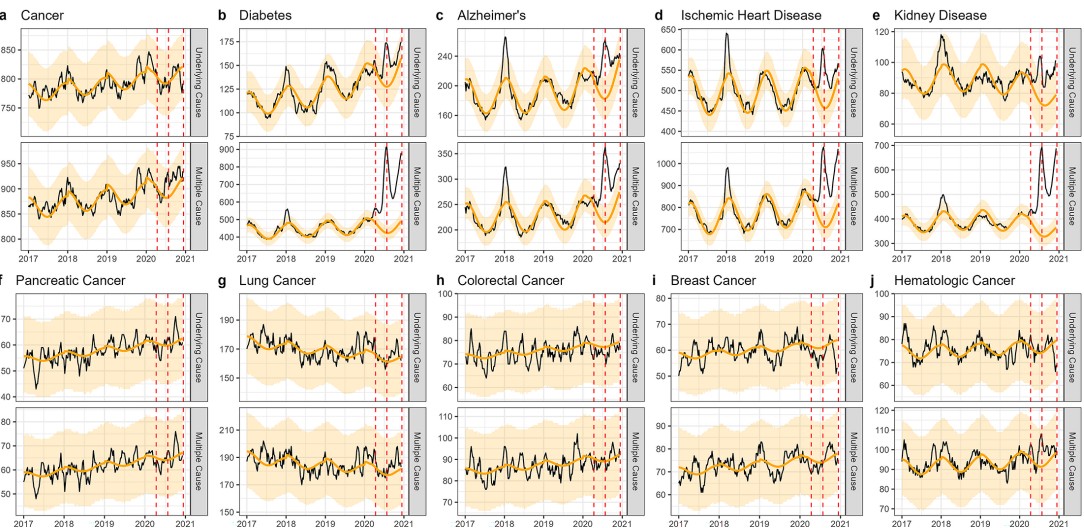

**Appendix 1—figure 7.** Weekly observed and estimated baseline mortality for each diagnosis group (Cancer (**a**), Diabetes (**b**), Alzheimer's (**c**), Ischemic Heart Disease (**d**), Kidney Disease (**e**), Pancreatic Cancer (**f**), Lung Cancer (**g**), Colorectal Cancer (**h**), Breast Cancer (**i**), Hematologica Cancer (**j**)) as both the underlying cause or anywhere on the death certificate (multiple cause) from 2017 to 2020 in Texas. Red dashed lines represent the timing of the peaks for the three COVID-19 waves in 2020. Baselines during the pandemic are projected based on the previous years of data.

**California**

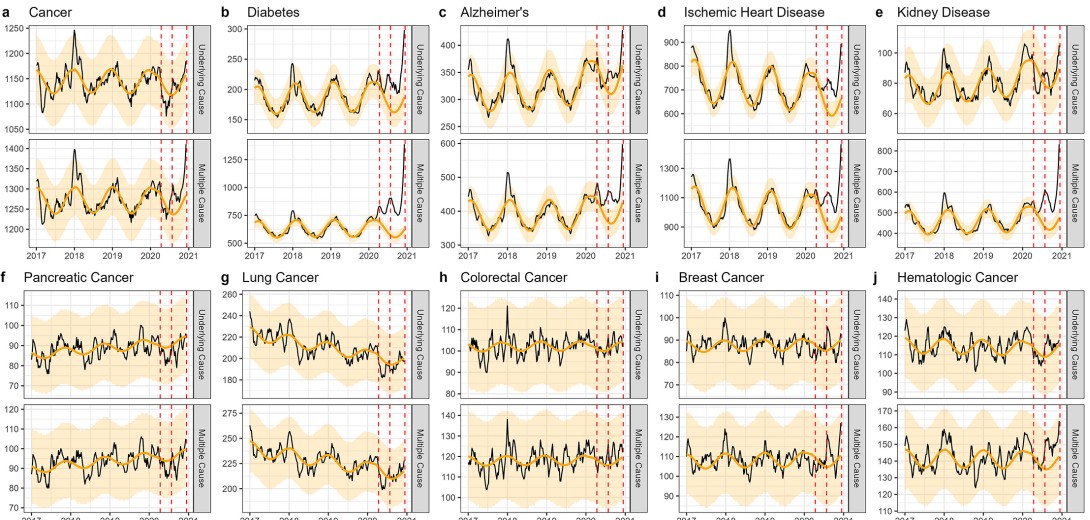

**Appendix 1—figure 8.** Weekly observed and estimated baseline mortality for each diagnosis group (Cancer (**a**), Diabetes (**b**), Alzheimer's (**c**), Ischemic Heart Disease (**d**), Kidney Disease (**e**), Pancreatic Cancer (**f**), Lung Cancer (**g**), Colorectal Cancer (**h**), Breast Cancer (**i**), Hematologica Cancer (**j**)) as both the underlying cause or anywhere on the death certificate (multiple cause) from 2017 to 2020 in New York. Red dashed lines represent the timing of the peaks for the three COVID-19 waves in 2020. Baselines during the pandemic are projected based on the previous years of data.

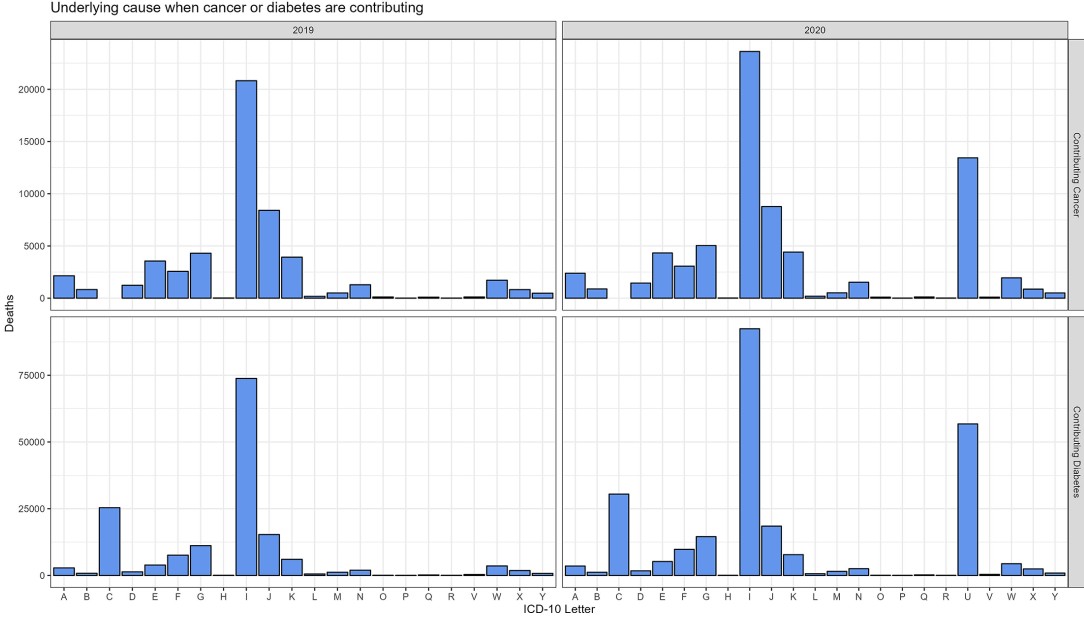

**Appendix 1—figure 9.** Comparison of ICD-10 letter categories between 2020 and 2019 for the underlying cause (UC) of death when cancer or diabetes are included on the death certificate, but are not listed as the UC of death. For both cancer and diabetes, I codes (diseases of the circulatory system) make up the majority of underlying deaths. The most notable difference between 2019 and 2020 is the increase in U codes, which includes COVID-19 (U071). In total there were 13,434 deaths ascribed to COVID-19 (UC deaths) among cancer multiple cause (MC) deaths. COVID-19 was included in <3% of all cancer deaths and 17% of diabetes deaths. In both cases it was listed as the UC on the majority of death certificates where it was included (81% and 97% for cancer and diabetes, respectively).

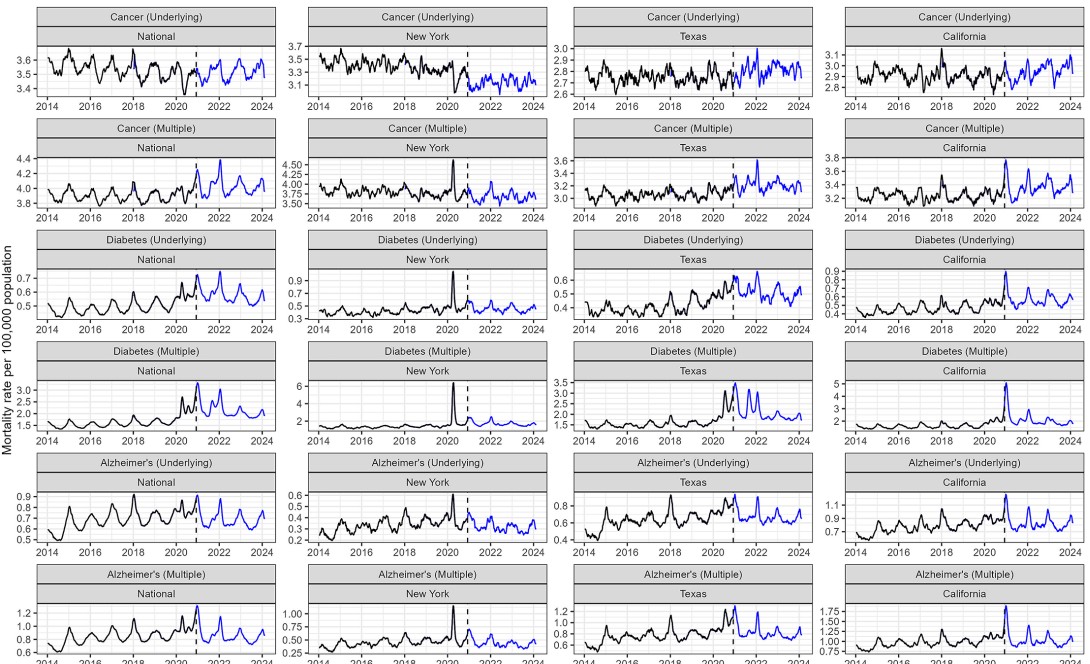

**Appendix 1—figure 10.** Post-2020 trends in cancer, diabetes, and Alzheimer's mortality. Aggregated weekly data was downloaded from CDC Wonder. Trends in cancer mortality rate appear stable in the national data and in Texas and California, but decreasing in New York. The diabetes mortality rate is higher post-2020 compared to earlier years across all states. Alzheimer's appears stable and slowly decreasing.

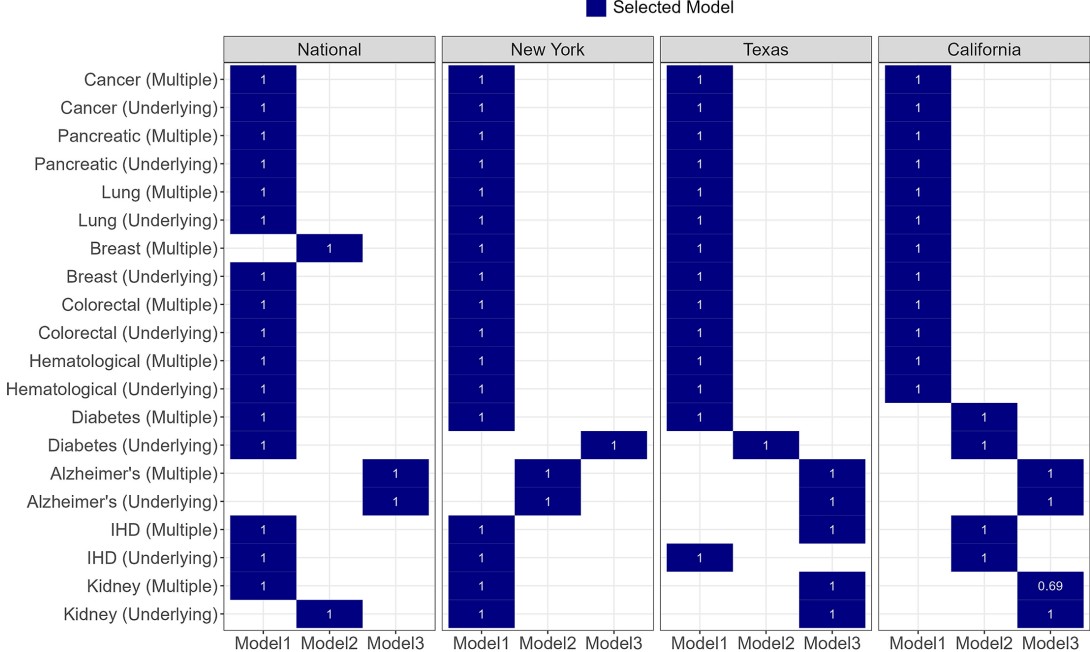

**Appendix 1—figure 11.** For each condition three time series models with different time trends were considered (see Materials and methods). The final model for each condition and location is indicated in blue. The final model was fit to 2014–2018 data only and used to predict the 2019 data. A coverage proportion (shown in white) was calculated as the proportion of observed 2019 data that fell within the projection intervals of the model. For all causes of death and states (except multiple cause [MC] kidney disease in California) the coverage proportion was 1, indicating that all data points fell within the prediction intervals.

