## [Editor Report · eLife assessment]

This **valuable** work explores death coding data to understand the impact of COVID-19 on cancer mortality. The work provides **solid** evidence that deaths with cancer as a contributing cause were not above what would be expected during pandemic waves, suggesting that cancer did not strongly increase the risk of dying of COVID-19. These results are an interesting exploration into the coding of causes of death that can be used to make sense of how deaths are coded during a pandemic in the presence of other underlying diseases, such as cancer.

---

## [Referee Report · Reviewer #1 (Public Review)]

Summary:

In the paper, the authors study whether the number of deaths in cancer patients in the USA went up or down during the first year (2020) of the COVID-19 pandemic. They found that the number of deaths with cancer mentioned on the death certificate went up, but only moderately. In fact, the excess with-cancer mortality was smaller than expected if cancer had no influence on the COVID mortality rate and all cancer patients got COVID with the same frequency as in the general population. The authors conclude that the data are consistent with cancer not being a risk factor for COVID and that cancer patients were likely actively shielding themselves from COVID infections.

Strengths:

The paper studies an important topic and uses sound statistical and modeling methodology. It analyzes both, deaths with cancer listed as the primary cause of death, as well as deaths with cancer listed as one of the contributing causes. The authors argue, correctly, that the latter is a more important and reliable indicator to study relationships between cancer and COVID. The authors supplement their US-wide analysis with analysing three states separately.

For comparison, the authors study excess mortality from diabetes and from Alzheimer's disease. They show that Covid-related excess mortality in these two groups of patients was expected to be much higher (than in cancer patients), and indeed that is what the data showed.

---

## [Author Response]

The following is the authors’ response to the original reviews.

**eLife assessment**
This valuable work explores death coding data to understand the impact of COVID-19 on cancer mortality. The work provides solid evidence that deaths with cancer as a contributing cause were not above what would be expected during pandemic waves, suggesting that cancer did not strongly increase the risk of dying of COVID-19. These results are an interesting exploration into the coding of causes of death that can be used to make sense of how deaths are coded during a pandemic in the presence of other underlying diseases, such as cancer.

We thank the editor and reviewers for the time they took to review our manuscript and for the thoughtful suggestions they provided. We have completed several revisions based on their feedback and we feel our paper is stronger as a result. However, none of these revisions change the overall conclusions of our study.

**Reviewer #1 (Public Review):**
Summary:In the paper "Disentangling the relationship between cancer mortality and COVID-19", the authors study whether the number of deaths in cancer patients in the USA went up or down during the first year (2020) of the COVID-19 pandemic. They found that the number of deaths with cancer mentioned on the death certificate went up, but only moderately. In fact, the excess with-cancer mortality was smaller than expected if cancer had no influence on the COVID mortality rate and all cancer patients got COVID with the same frequency as in the general population. The authors conclude that the data show no evidence of cancer being a risk factor for COVID and that the cancer patients were likely actively shielding themselves from COVID infections.Strengths:The paper studies an important topic and uses sound statistical and modeling methodology. It analyzes both, deaths with cancer listed as the primary cause of death, as well as deaths with cancer listed as one of the contributing causes. The authors argue, correctly, that the latter is a more important and reliable indicator to study relationships between cancer and COVID. The authors supplement their US-wide analysis by analysing three states separately.Weaknesses:The main findings of the paper can be summarized as six numbers. Nationally, in 2022, multiple-cause cancer deaths went up by 2%, Alzheimer's deaths by 31%, and diabetes deaths by 39%. At the same time, assuming no relationship between these diseases and either Covid infection risk or Covid mortality risk, the deaths should have gone up by 7%, 46%, and 28%. The authors focus on cancer deaths and as 2% < 7%, conclude that cancer is not a risk factor for COVID and that cancer patients must have "shielded" themselves against Covid infections.However, I did not find any discussion of the other two diseases. For diabetes, the observed excess was 39% instead of "predicted by the null model" 28%. I assume this should be interpreted as diabetes being a risk factor for Covid deaths. I think this should be spelled out, and also compared to existing estimates of increased Covid IFR associated with diabetes.And what about Alzheimer's? Why was the observed excess 31% vs the predicted 46%? Is this also a shielding effect? Does the spring wave in NY provide some evidence here? Why/how would Alzheimer's patients be shielded? In any case, this needs to be discussed and currently, it is not.

We thank the reviewer for their positive feedback on the paper and for these suggestions. It is true that we have emphasized the impact on cancer deaths, as this was the primary aim of the paper. In the revised version, we have expanded the results and discussion sections to more fully describe the other chronic conditions we used as comparators (lines 267-284;346 – 386).

Note that we are somewhat reluctant to designate any of these conditions as risk factors based solely on comparing the time series model with the demographic model of our expectations. As we mention in the discussion, there is considerable uncertainty around estimates from the demographic model in terms of the size of the population-at-risk, the mean age of the population-at-risk, and the COVID-19 infection rates and infection fatality ratios. Our demographic model is primarily used to demonstrate the effects of competing risks across types of cancers and chronic conditions, since these findings are robust to model assumptions. In contrast, the demographic model should be used with caution if the goal is to titrate the level of these risk factors (as the level of imputed risk is dependent on model assumptions). In the updated version of the manuscript, we have included uncertainty intervals in Table 3, using the upper and lower bounds of the estimated infection rates and IFRs, to better represent this uncertainty. We have also discussed this uncertainty more explicitly in the text and ran sensitivity analyses with different infection rate assumptions in the discussion (lines 354-362; 367 -370).

We would like to note that rather than interpreting the absolute results, we used this demographic model as a tool to understand the relative differences between these conditions. From the demographic model we determined that we would expect to see much higher mortality in diabetes and Alzheimer’s deaths compared to cancer deaths due to three factors (1. Size of population-at-risk, 2. Mean age of the population-at-risk, 3. Baseline risk of mortality from the condition), that are separate from the COVID-19 associated IFR. And in general, this is what we observed.

In comparing the results from the demographic model to the observed excess, diabetes does standout as an outlier from cancer and Alzheimer’s disease in that the observed excess is consistently above the null hypothesis which does lend support to the conclusion that diabetes is in fact a risk factor for COVID-19. A conclusion which is also supported by many other studies. Our findings for hematological cancers are also similar, in that we find consistent support for this condition being a risk factor. We have commented on this in the discussion and added a few references (lines 346-354; 395-403).

Our hypothesis regarding non-hematological cancer deaths (lower than anticipated mortality due to shielding) could also apply to Alzheimer’s deaths. Furthermore, we used the COVID-19 attack rate for individuals >65 years (based on the data that is available), but we estimate that the mean age of Alzheimer’s patients is actually 80-81 years, so this attack rate may in fact be a bit too high, which would increase our expected excess. We have commented on this in the discussion (lines 363-377).

**Reviewer #2 (Public Review):**
The article is very well written, and the approach is quite novel. I have two major methodological comments, that if addressed will add to the robustness of the results.(1) Model for estimating expected mortality. There is a large literature using a different model to predict expected mortality during the pandemic. Different models come with different caveats, see the example of the WHO estimates in Germany and the performance of splines (Msemburi et al Nature 2023 and Ferenci BMC Medical Research Methodology 2023). In addition, it is a common practice to include covariates to help the predictions (e.g., temperature and national holidays, see Kontis et al Nature Medicine 2020). Last, fitting the model-independent for each region, neglects potential correlation patterns in the neighbouring regions, see Blangiardo et al 2020 PlosONE.

Thank you for these comments and suggestions. We agree there are a range of methods that can be used for this type of analysis, and they all come with their strengths, weaknesses, and caveats. Broadly, the approach we chose was to fit the data before the pandemic (2014-2019), and project forward into 2020. To our knowledge it is not a best practice to use an interpolating spline function to extrapolate to future years. This is demonstrated by the WHO estimates in Germany in the paper you mention. This was our motivation for using polynomial and harmonic terms.

Based on the above:a. I believe that the authors need to run a cross-validation to justify model performance. I would suggest training the data leaving out the last year for which they have mortality and assessing how the model predicts forward. Important metrics for the prediction performance include mean square error and coverage probability, see Konstantinoudis et al Nature Communications 2023. The authors need to provide metrics for all regions and health outcomes.

Thank you for this suggestion. We agree that our paper could be strengthened by including cross validation metrics to justify model performance. Based on this suggestion, and your observations regarding Alzheimer’s disease, we have done two things. First, for the full pre-pandemic period (2014-2019) for each chronic condition and location we tested three different models with different degree polynomials (1. linear only, 2. linear + second degree polynomial, 3. linear + second degree polynomial + third degree polynomial) and used AIC to select the best model for each condition and location. Next, also in response to your suggestion, we estimated coverage statistics. Using the best fit model from the previous step, we then fit the model to data from 2014-2018 only and used the model to predict the 2019 data. We calculated the coverage probability as the proportion of weekly observed data points that fell within the 95% prediction interval. For all causes of death and locations the coverage probability was 100% (with the exception of multiple cause kidney disease in California, which is only shown in the appendix). The methods and results have been updated to reflect this change and we have added a figure to the appendix showing the selected model and coverage probability for each cause of death and location (lines 504 – 519; 847-859; Appendix 1- Figure 11).

b. In the context of validating the estimates, I think the authors need to carefully address the Alzheimer case, see Figure 2. It seems that the long-term trends pick an inverse U-shape relationship which could be an overfit. In general, polynomials tend to overfit (in this case the authors use a polynomial of second degree).It would be interesting to see how the results change if they also include a cubic term in a sensitivity analysis.

Thank you for this observation. Based on the changes described above, the model for Alzheimer’s disease now includes a cubic term in the national data and in Texas and California. The model with the second-degree polynomial remained the best fit for New York (Appendix 1 – Figure 11).

c. The authors can help with the predictions using temperature and national holidays, but if they show in the cross-validation that the model performs adequately, this would be fine.

At the scale of the US, adding temperature or environmental covariates is difficult and few US-wide models do so (see Goldstein 2012 and Quandelacy 2014 for examples from influenza). Furthermore, because we are looking at chronic disease outcomes, it is unclear that viral covariates or national holidays would drive these outcomes in the same way as they would if we were looking at mortality outcomes more directly related to transmissible diseases (such as respiratory mortality). Our cross validation also indicates that our models fit well without these additional covariates.

d. It would be nice to see a model across the US, accounting for geography and spatial correlation. If the authors don't want to fit conditional autoregressive models in the Bayesian framework, they could just use a random intercept per region.

We think the reviewer is mistaken here about the scale of our national analysis. Our national analysis did not fit independent models for each state or region. Rather, we fit a single model to the weekly-level national mortality data where counts for the whole of the US have been aggregated. We have clarified in the text (lines 156, 464). As such, we do not feel a model accounting for spatial correlation would be appropriate nor would we be able to include a random intercept for each region. We did fit three states independently (NY, TX, CA), but these states are very geographically distant from each other and unlikely to be correlated. These states were chosen in part because of their large population sizes, yet even in these states, confidence intervals were very wide for certain causes of death. Fitting models to each of the 50 US states, most of which are smaller than those chosen here, would exacerbate this issue.

(2) I think the demographic model needs further elaboration. It would be nice to show more details, the mathematical formula of this model in the supplement, and explain the assumptions

Thank you for this comment. We have added additional details on the demographic model to the methods. We have also extended this analysis to each state to further strengthen our conclusions (lines 548-590).

Reviewing Editor Recommendations:I think that perhaps something that is missing is that the authors never make their underlying assumption explicit: they are assuming that if cancer increases the risk of dying of COVID-19, this would be reflected in the data on multiple causes of death where cancer would be listed as one of the multiple causes rather than as the underlying cause, and that their conclusions are predicated on this assumption. I would suggest explicitly stating this assumption, as opposed to other reasons why cancer mortality would increase (ex. if cancer care worsened during pandemic waves leading to poorer cancer survival).

Response: Thank you for this suggestion. We have added a few sentences to the introduction to make this assumption clear (lines 106-112).

**Reviewer #1 (Recommendations For The Authors):**
- It could make sense to add "in the United States" into the title, as the paper only analyses US data.- It may make sense to reformulate the title from "disentangling the relationship..." into something that conveys the actual findings, e.g. "Lack of excess cancer mortality during Covid-19 pandemic" or something similar. Currently, the title tells nothing about the findings.

Thank you for these suggestions. We have added “in the US” to the title. However, we feel that our findings are a bit more subtle than the suggested reformulation would imply, and we prefer to leave it in its current form.

- Abstract, lines 42--45: This is the main finding of the paper, but I feel it is simplified too strongly in the abstract. Your simulations do *not* "largely explain" excess mortality with cancer; they give higher numbers! Which you interpret as "shielding" etc., but this is completely absent from the abstract. This sentence makes the impression that you got a good fit between simulated excess and real excess, which I would say is not the case.

Thank you for this comment. We have rephrased the sentence in the abstract to better reflect our intentions for using the demographic model (lines 46-49). As stated above, the purpose of the demographic model was not to give a good fit with the observed excess mortality. Rather, we used the demographic model as a tool to understand the relative differences between these conditions in terms of expected excess mortality given the size, age-distribution, and underlying risk of death from the condition itself, assuming similar IFR and attack rates. And based on this, we conclude that it is not necessarily surprising that we see higher excess mortality for diabetes and Alzheimer’s compared to cancer.

- Results line 237: you write that it's "more consistent with the null hypothesis", however clearly it is *not* consistent with the null hypothesis either (because 2% < 7%). You discuss in the Discussion that it may be due to shielding, but it would be good to have at least one sentence about it already here in the Results, and refer to the Discussion.

We have mentioned this in the results and refer to the discussion (lines 277-278).

- Results line 239: why was it closer to the assumption of relative risk 2? If I understand correctly, your model prediction for risk=1 was 7% and for risk=2 it was 13%. In NY you observed 8% (line 187). How is this closer to risk=2?

Thank you for this observation. We have updated the demographic model with new data, extended the model to state-level data, and included confidence intervals on these estimates. We have also added additional discussion around the differences between our observations and expectations (lines 249-284).

- Discussion line 275: "we did not expect to see large increases" -- why exactly? Please spell it out here. Was it due to the age distribution of the cancer patients? Was it due to the high cancer death risk?

We demonstrate that it is the higher baseline risk of death for cancer that seems to be driving our low expectations for cancer excess mortality (lines 304-320). We have added this to the sentence to clarify our conclusions on this point and have added a figure to better illustrate this concept of competing risks (Figure 6).

- Methods, line 405: perhaps it makes sense to cite some other notable papers on Covid excess mortality such as Msemburi et al Nature 2023, Karlinsky & Kobak eLife 2021, Islam et al BMJ 2021, etc.

Thank you for mentioning this oversight. We certainly should have cited these papers and have included them in the updated version.

- Methods line 410: why did you use a 5-week moving average? Why not fit raw weekly death counts? NB regression should be able to deal with it.

Smoothing time series data with a moving average prior to running regression models is a very common practice. We did a sensitivity analysis using the raw data. This produced excess estimates with slightly larger confidence intervals, but does not change the overall conclusions of the paper.

- Methods line 416: please indicate the software/library/package you used for fitting NB regression.

We fit the NB regression using the MASS package in R version 4.3. We have added this to the methods (line 519).

- Line 489: ORCHID -> ORCID